# Vitamin B5 metabolism is essential for vacuolar and mitochondrial functions and drug detoxification in fungi
Jae-Yeon Choi [1], Shalev Gihaz [1], Muhammad Munshi [1], Pallavi Singh [1], Pratap Vydyam[1], Patrice Hamel[2], Emily M. Adams [3,4], Xinghui Sun[5], Oleh Khalimonchuk[5,6,7], Kevin Fuller[3,4] & Choukri Ben Mamoun [1] ✉

Fungal infections, a leading cause of mortality among eukaryotic pathogens, pose a growing global health threat due to the rise of drug-resistant strains. New therapeutic strategies are urgently needed to combat this challenge. The PCA pathway for biosynthesis of Co-enzyme A (CoA) and Acetyl-CoA (AcCoA) from vitamin B5 (pantothenic acid) has been validated as an excellent target for the development of new antimicrobials against fungi and protozoa. The pathway regulates key cellular processes including metabolism of fatty acids, amino acids, sterols, and heme. In this study, we provide genetic evidence that disruption of the PCA pathway in *Saccharomyces cerevisiae* results in a significant alteration in the susceptibility of fungi to a wide range of xenobiotics, including clinically approved antifungal drugs through alteration of vacuolar morphology and drug detoxification. The drug potentiation mediated by genetic regulation of genes in the PCA pathway could be recapitulated using the pantazine analog PZ-2891 as well as the celecoxib derivative, AR-12 through inhibition of fungal AcCoA synthase activity. Collectively, the data validate the PCA pathway as a suitable target for enhancing the efficacy and safety of current antifungal therapies.

Annually, invasive fungal infections kill over 1.6 million people globally. A 2023 report by the US Center for Disease Control (CDC) warned that the rapid growth of incidence of infection by multi-drug resistant *Candida auris* urgently threatens domestic and global public health[1]. Azole-resistant *Aspergillus* species[2,3] (classified by the CDC as an emerging threat) and other drug-resistant strains[4] contribute to the urgency for development of new antifungal therapeutic strategies.

Development of highly selective and potent antifungal drugs (AFDs) is hampered by the similarities shared between mammalian and fungal cells and the latter's drug resistance mechanisms[5–7]. One of these commonalities lies in the central importance of the synthesis of Coenzyme-A (CoA), an obligate cofactor of approximately 4–9% of all known enzymes[8] and a precursor for acetyl-CoA (AcCoA), an acetyl carrier essential for the operation of synthetic and oxidative pathways. The biosynthesis of CoA involves a 5-step enzymatic process that begins with the phosphorylation of vitamin B5 (pantothenate or PA) by pantothenate kinases (PanKs) to form

4-phosphopantothenate. Various components of the PA-CoA-AcCoA (PCA) pathway (Fig. 1a) have previously been explored as potential druggable targets for antimicrobials, including against malaria parasites[9,10]. Human cells express four pantothenate kinases (PanK1α, PanK1β, PanK2, and PanK3) in different tissues. Mutations in the *PANK2* gene, encoding a mitochondrial PanK enzyme, have been linked to a debilitating pediatric neurodegenerative disease named pantothenate-kinase associated neurodegeneration (PKAN)[11]. Modulation of the cytoplasmic PanK3 activity using small molecule modulators to compensate for the loss of PanK2 has become of interest in the treatment of PKAN. In contrast, in *Saccharomyces cerevisiae* as well as most causative agents of invasive fungal infections, a single copy of the PanK enzyme performs the first step in the PCA pathway[12,13]. Fungal *PANK* genes have all been shown to be essential for cell viability as complete or conditional knockouts of these genes lead to cell death[12–16]. As a result, efforts to identify inhibitors of fungal PanK enzymes has been considered an attractive strategy for the development of new

[1]Section of Infectious Diseases, Department of Medicine, Department of Microbial Pathogenesis, Yale University School of Medicine, New Haven, CT, USA. [2]Departments of Molecular Genetics and Department of Biological Chemistry and Pharmacology, The Ohio State University, Columbus, OH, USA. [3]Department of Microbiology and Immunology, University of Oklahoma Health Sciences Center, Oklahoma City, OK, USA. [4]Department of Ophthalmology, University of Oklahoma Health Sciences Center, Oklahoma City, OK, USA. [5]Department of Biochemistry, University of Nebraska-Lincoln, Lincoln, NE, USA. [6]Nebraska Redox Biology Center, Lincoln, NE, USA. [7]Fred & Pamela Buffett Cancer Center, Omaha, NE, USA. ✉e-mail: choukri.benmamoun@yale.edu

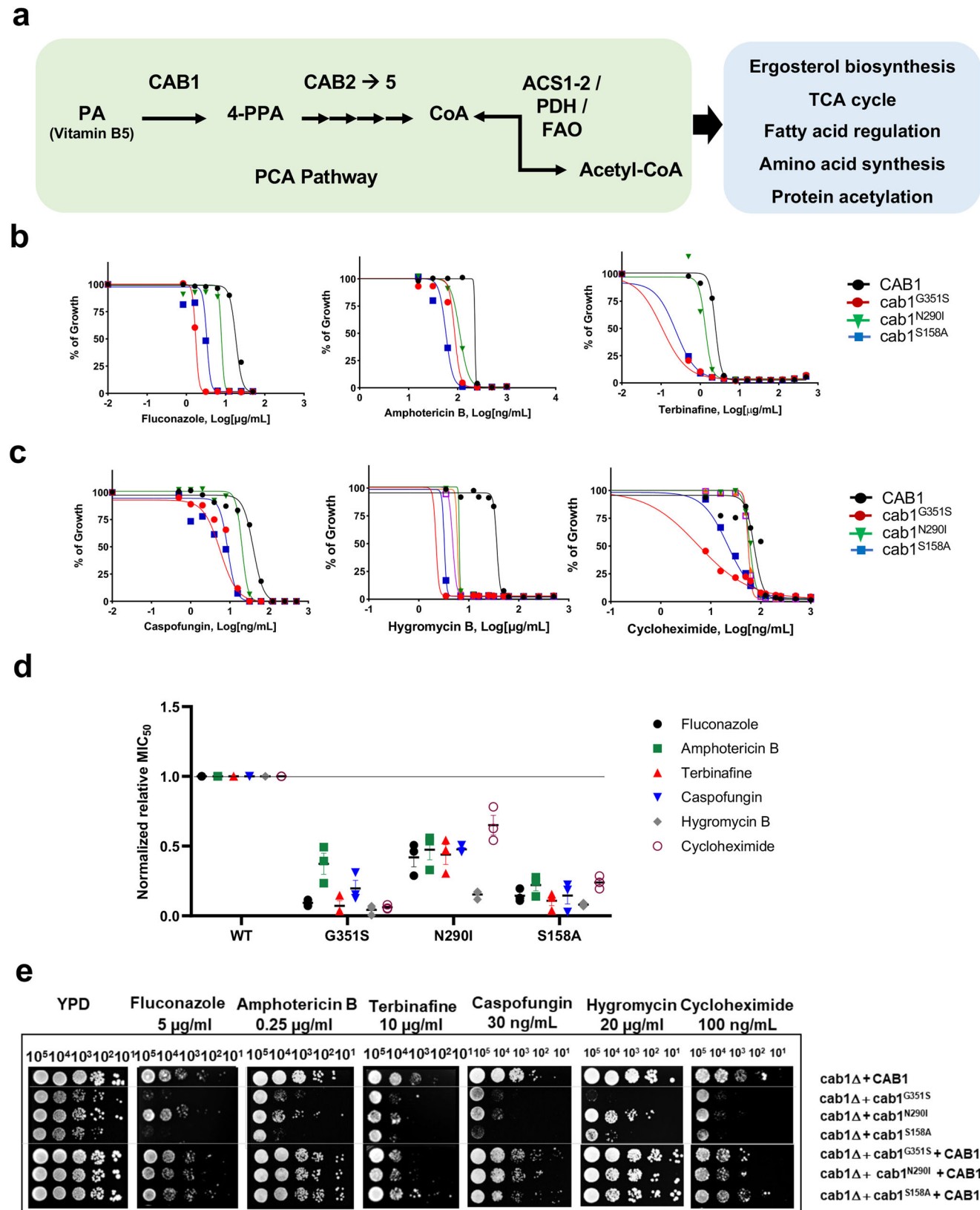

antifungal drugs. A high-throughput screen of a library of 156,593 chemical compounds against the *Aspergillus fumigatus* PanK enzyme (AfPanK) identified several pyrimidone triazoles as potential strong and selective inhibitors of AfPanK activity[16]. The first crystal structure of a fungal PanK enzyme was obtained at 1.8 Å resolution for the *S. cerevisiae*

PanK enzyme, Cab1, encoded by the *CAB1* gene[16]. It consists of two monomers sharing a dimeric interface that accommodates the ATP-binding pocket, active site, and part of the PA binding domain. The structure of the enzyme was also determined in the presence of pyrimidone triazoles inhibitors[16,17].

**Fig. 1 | *Cab1* mutants display growth defects and increased susceptibility toward ergosterol targeting drugs. a** Schematic diagram of Coenzyme A synthetic pathway and biological roles of the Acetyl-CoA. Pantothenate kinase (PanK) catalyzes the phosphorylation of pantothenic acid to form 4'-phosphopantothenic acid, the first step in the biosynthesis of CoA. Phosphopantothenoylcysteine synthase (PPCS) converts 4'-phosphopantothenic acid to phosphopantothenoylcysteine. Phospho-pantothenoylcysteine decarboxylase (PPCD) catalyzes the decarboxylation of phosphopantothenoylcysteine to form pantetheine. Pantetheine kinase (PanK) catalyzes the phosphorylation of pantetheine to form dephospho-CoA. Dephospho-CoA kinase (DPCK) catalyzes the transfer of a phosphate group from ATP to dephospho-CoA, resulting in the formation of CoA. **b- c** The *cab1* mutants display susceptibility toward drugs targeting ergosterol pathway and non-ergosterol tar-geting pathway. Yeast liquid growth curve assay with known AFDs targeting the ergosterol biosynthesis pathway was done using yeast cells harboring different *CAB1* mutants. Cells were inoculated into 100 μL of YPD liquid media containing the antifungals in serial dilutions at 10 cells per μL ratio and incubated at 30 °C while cell growth was monitored by $OD_{600}$. % of cell growth of individual mutant strain in presence of AFDs was obtained compared to the cell growth in the vehicle control. Liquid growth assays were conducted in 3 replicates (n = 3) and the plotted graphs represent the average of 3 data sets. **d** Normalized relative $MIC_{50}$ of yeast strains expressing *Cab1* variants against AFDs. $MIC_{50}$ values were determined on $MIC_{50}$ data obtained from the liquid growth assays presented in Fig. 1b, c. These values were calculated by dividing the $MIC_{50}$ value of the *cab1* mutant or the wild type strain for the specified AFD by the $MIC_{50}$ value of the wild type strain. The plotted graph represents average of four data sets ± SD. **e** Complementation of the wild type *CAB1* gene restores AFD resistance in the *cab1* mutants to the similar level as that observed in wild-type cells. The *cab1Δ/w303-1B* strain harboring 1) wild type *CAB1*, 2) *cab1^mutant*, or 3) *cab1^mutant* + wild type *CAB1*, were inoculated into synthetic glucose medium and grown overnight. Cells were harvested and re-suspended in 0.9% NaCl. Ten-fold serial dilutions of cells were spotted onto the YPD plates containing flu-conazole (5 μg/mL), amphotericin B (0.25 μg/mL), and terbinafine (10 μg/mL), caspofungin (30 ng/mL), hygromycin B (20 μg/mL), or cycloheximide (100 ng/mL), and incubated at 30 °C for 3 days. The representative images are from two inde-pendent experiments, each performed in duplicates.

The first evidence linking PanK activity to drug susceptibility was reported by Chiu et al. using a *S. cerevisiae cab1^ts* thermosensitive mutant carrying a mutated chromosomal *CAB1* gene[12]. The *cab1^ts* thermosensitive strain (JS91.14-24) was generated following EMS mutagenesis to select mutants altered in saturated fatty acid biosynthesis[13,18]. To gain further insights into the link between the PCA pathway and antifungal drug sus-ceptibility and to unravel the underlying molecular mechanisms, we used strains that carry a chromosomal deletion of *CAB1* but carry either the wild type *CAB1* or mutated alleles *cab1^G351S*, *cab1^N290I* and *cab1^S158A* on a cen-tromeric plasmid[14–16]. All three mutations cause significant decrease in Cab1 pantothenate kinase activity and kinetic properties[16]. The G351 residue is positioned on a helical loop near crucial catalytic residues with substitutions to either alanine or serine resulting in major alterations in the kinetic properties of the enzyme[16]; the N290I mutation mirrors a mutation in the human *PANK2* gene and associated with PKAN[15,17]; whereas the S158 residue is in the Cab1 catalytic site and its substitution to alanine results in a dramatic decrease in enzyme activity[16]. Our data demonstrate that these mutations render yeast cells highly susceptible to a broad-spectrum of antifungals, including those that target ergosterol biosynthesis, protein synthesis, cell wall formation, and RNA synthesis. Our cell biological and pharmacological analyses established a role for the PCA pathway in the regulation of vacuolar function and drug detoxification. We identified the PanK3 orthosteric activator, PZ-2891, and the celecoxib derivative, AR-12, as modulators of the PCA pathway and enhancers of fungal susceptibility to antifungal drugs.

## Results
### PanK mutants have broad-range susceptibility to antifungal drugs
Previous studies in *S. cerevisiae* and *A. fumigatus* have shown that reduced PanK activity caused by substitution of glycine 351 to serine (*cab1^G351S*) in the *S. cerevisiae* Cab1 enzyme, or reduced expression, using a conditional promoter to drive the expression of the *A. fumigatus* AfPanK, result in altered susceptibility to amphotericin B and voriconazole, respectively[12,16]. To gain further insights into the link between Cab1 activity, the PCA pathway (Fig. 1a), and fungal susceptibility to antifungal drugs, a detailed characterization of *S. cerevisiae* drug susceptibility using strains carrying various mutations in the *CAB1* gene was conducted. The yeast strains used in these studies carry a chromosomal deletion of the *CAB1* gene but express on a centromeric plasmid either a wild type *CAB1* (*cab1Δ + CAB1*), mutant alleles of *CAB1* (*cab1Δ + cab1^m* with m = *cab1^G351S*, *cab1^N290I*, and *cab1^S158A*) or both the mutant alleles and the wild type *CAB1* (add-back: *cab1Δ + cab1^m + CAB1*). As shown Fig. 1b, all the yeast strains expressing different *CAB1* alleles exhibited increased susceptibility to fluconazole, amphotericin B, and terbinafine, compared to the isogenic strain carrying the wild type *CAB1*. The *cab1Δ + cab1^G351S* mutant was found to be highly susceptible to all ergosterol biosynthesis inhibitors examined with a reduction in $MIC_{50}$

values determined to be ~12, ~15, and ~3-fold for fluconazole, terbinafine, and amphotericin, respectively, compared to the wild type strains (Fig. 1d). Similarly, the *cab1Δ+cab1^S158A* mutant displayed reduced $MIC_{50}$ values with fold reductions compared to the wild type determined to be ~7, ~9, and ~5 for fluconazole, terbinafine, and amphotericin, respectively (Fig. 1d).

The broad-range susceptibility of *cab1* mutants to antifungal drugs led us to examine whether the underlying mechanism could be linked to dis-ruption of a particular metabolic process such as ergosterol biosynthesis by the PCA pathway or to a broader disruption of yeast's ability to detoxify xenobiotics. Therefore, we examined the susceptibility of the mutants to drugs that target unrelated pathways including hygromycin B and cyclo-heximide, which target protein synthesis, and caspofungin, which targets cell wall integrity (Fig. 1c). Similar to their susceptibility to ergosterol bio-synthesis inhibitors, the *cab1* mutant alleles showed higher susceptibility to caspofungin, hygromycin B and cycloheximide compared to the wild type (Fig. 1c). The *cab1^G351S* mutation resulted in the highest drug susceptibility with the $MIC_{50}$ values of caspofungin, hygromycin B, and cycloheximide determined to be ~5, ~23, and ~16- fold lower compared to the wild type (Fig. 1d). Similarly, *cab1^S158A* mutation resulted in reduced $MIC_{50}$ values for caspofungin, hygromycin B, and cycloheximide by ~6, ~13, and ~4- fold compared to the wild type (Fig. 1d). All complemented strains carrying the wild-type *CAB1* gene displayed susceptibility levels comparable to those of the wild-type (WT) strain (Fig. 1e). Altogether these data demonstrate that inhibition of PanK activity leads to enhanced susceptibility to a wide variety of antifungal drugs.

### PanK-deficient cells have altered vacuole biogenesis and xeno-biotic detoxification mechanism
The overall enhanced drug susceptibility of Cab1 defective mutants led us to investigate a possible role of the vacuole in this mechanism. In fungi, the vacuole plays a critical role in the detoxification of xenobiotics such as drugs and metals[12,19]. Therefore, we examined whether the *cab1* mutants might also be susceptible to metals such as $FeSO_4$ and $CuSO_4$, which are detoxified in the vacuole. Consistent with an altered vacuolar detoxification in the mutants, the growth of the *cab1Δ+cab1^G351S*, *cab1Δ+cab1^N290I* and *cab1Δ+cab1^S158A* mutants was reduced in media supplemented with $FeSO_4$ or $CuSO_4$ compared to the wild type and complemented strains (Fig. 2a). Vacuolar morphology was further investigated by measuring the accumu-lation of the cell-tracker dye, CMAC, using fluorescence microscopy[19,20]. The *cab1Δ+cab1^G351S*, *cab1Δ+cab1^N290I* and *cab1Δ+cab1^S158A* mutants were all found to have enlarged vacuoles compared to the wild type and com-plemented strains (Fig. 2b, c), with the vacuoles in the mutants occupying 60 to 70% more of the total cell area compared to the vacuoles of the wild-type and complemented strains. Vacuolar enlargement in the mutants was fur-ther confirmed by electron microscopy as in Fig. 2d. Taken together, these data provide the first evidence that pantothenate phosphorylation regulates vacuolar homeostasis and xenobiotic detoxification.

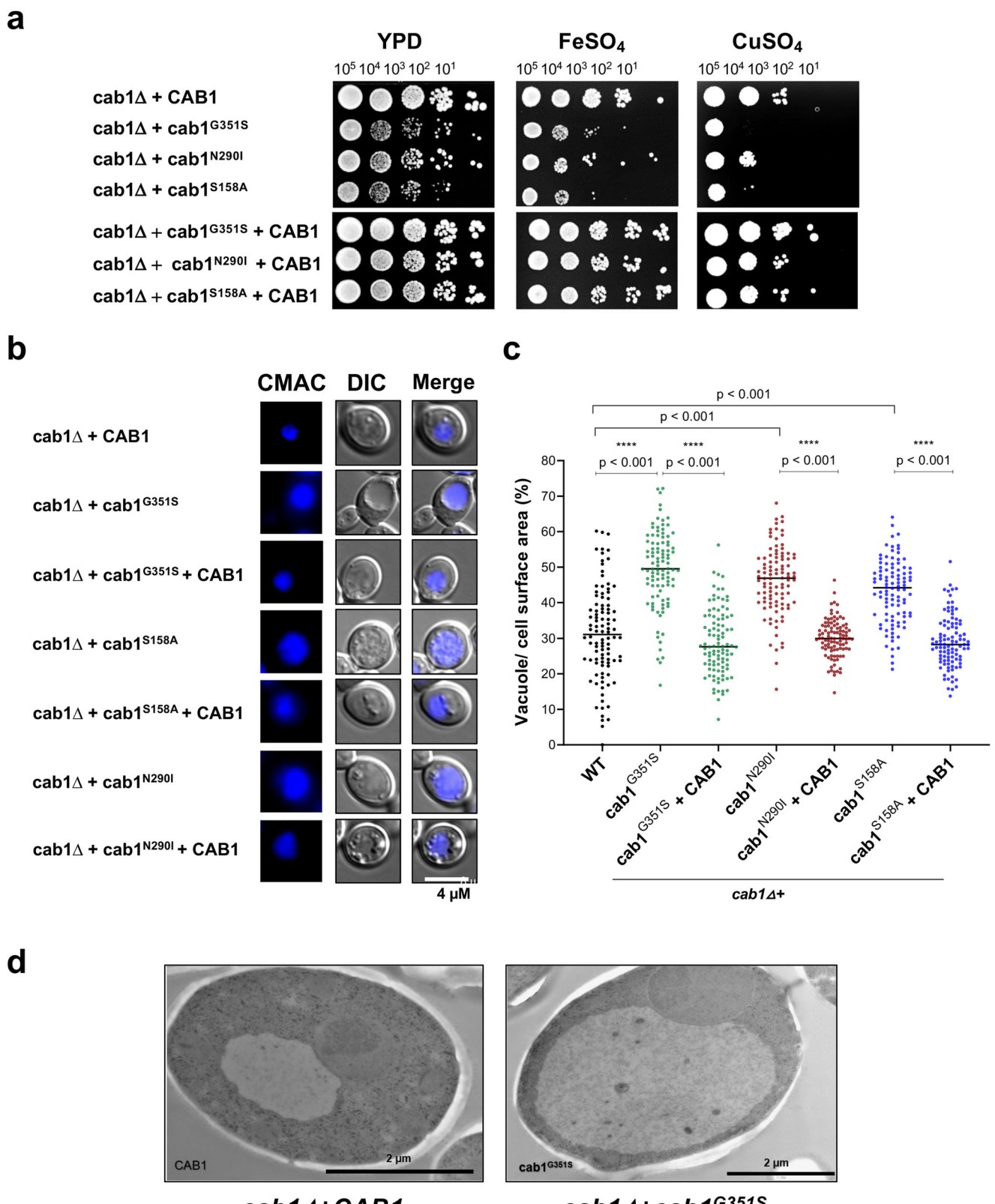

Recent studies have established an important role for vacuolar biogenesis in the maintenance of mitochondrial function and integrity[21,22]. Therefore, we surmised that altered vacuolar function in the *cab1* mutants could also result in altered mitochondrial function. Accordingly, the *cab1Δ +cab1$^{G351S}$, cab1Δ+cab1$^{N290I}$* and *cab1Δ+cab1$^{S158A}$* mutants showed severe growth defects on non-fermentable carbon sources (glycerol, ethanol, and lactate-based media) (Fig. 3a) and altered oxygen consumption rates (OCR) (Fig. 3b–e). Consistent with these findings, immunofluorescence assays aimed to localize the mitochondrial outer membrane protein Por1p revealed that unlike the wild-type and complemented strains, the *cab1* mutants exhibited fragmented mitochondria (Fig. 3f). Since the overproduction of reactive oxygen species (ROS) is associated with dysfunctional

**Fig. 2 | Yeast strains expressing various *CAB1* mutations have defects in detoxification and vacuolar function and structure. a** The *cab1* mutations altered yeast capacity to overcome metal toxicity. Solid growth assays were performed with the yeast strains described above in glucose media supplemented with 7 mM FeSO$_4$ or with 10 mM CuSO$_4$. The representative images are from two independent experiments, each performed in duplicates. **b** Morphological analysis of vacuolar defects shows that *cab1* mutant strains have unusually enlarged vacuoles. Immunofluorescence microscopy images of *cab1Δ + cab1*$^{mutant}$ strains described above show enlarged and/or fragmented vacuoles, while the *cab1Δ + CAB1*$^{WT}$ and the *cab1Δ +*

*cab1*$^{mutant}$ *+ CAB1*$^{WT}$ strains have typical vacuolar morphology. The representative images are from two independent experiments, each performed in duplicates. **c** Quantitative analysis of vacuolar area size (as a proportion of total cell area) as a function of *CAB1* status; n = 100 cells per condition. Statistical significance was determined using t-test (*p = 0.05*) with GraphPad Prism and the corresponding *p*-values (***p < 0.001) are indicated. **d** Electron microscopy images of *cab1Δ + CAB1*$^{WT}$ and *cab1Δ+cab1*$^{G351S}$ single cells confirm enlargement of vacuole induced by G351S mutation. The representative EM images are from three biological samples, with minimum of 100 cells analyzed per condition.

mitochondria[23,24], cellular ROS levels of the *cab1* mutants were also determined by measuring the conversion of the non-fluorescent dihydrorhodamine 123 (DHR-123) to the fluorescent rhodamine 123. As shown in Fig. 3g, ROS levels in the *cab1Δ+cab1*$^{G351S}$, *cab1Δ+cab1*$^{N290I}$ and *cab1Δ +cab1*$^{S158A}$ mutants were found to be 10.1, 13.0 and 4.1-fold higher than in the wild-type and complemented strains, respectively.

### Altered pantothenate phosphorylation leads to reduced pantothenate utilization and CoA biosynthesis and increased cysteine levels in yeast

To gain further insights into the mechanism by which altered pantothenate phosphorylation leads to defective vacuolar biogenesis, we assessed the effect of the *CAB1* mutations on the activity of the PCA pathway. Endogenous pantothenate kinase activity of *cab1Δ+cab1*$^{G351S}$, *cab1Δ+cab1*$^{S158A}$, and *cab1Δ+cab1*$^{N290I}$ was assessed by monitoring the phosphorylation and subsequent utilization of $^{14}$C-pantothenate (Fig. 4a). As shown in Fig. 4a, all *cab1* mutants showed significantly lower levels of $^{14}$C-pantothenate utilization compared to the isogenic strain carrying the wild type *CAB1*. The lowest PanK activity (~2% that of the wild type) was measured for the *cab1*$^{S158A}$ mutant followed by 25% for the *cab1*$^{G351S}$ mutant and 34% for the *cab1*$^{N290I}$ mutant. Expression of the wild-type *CAB1* gene in these strains restored PA utilization to levels similar or above those in the isogenic wild type-strain. Consistent with the reduced pantothenate utilization in the mutants, cellular CoA levels in the mutants were also significantly lower compared to the wild-type and complemented strains (Figs. 4b and S1). Because reduced pantothenate phosphorylation results in less phosphopantothenate available for the second step in CoA biosynthesis catalyzed by phosphopantothenoylcysteine synthetase (Cab2) to form phosphopantothenoylcysteine from phosphopantothenate and cysteine (Fig. 4d), we reasoned that altered Cab1 activity would also result in accumulation of cysteine. As shown in Fig. 4c, cysteine levels increased by 2.7, 3.3, and 2.2-fold in the *cab1*$^{G351S}$, *cab1*$^{S158A}$, and *cab1*$^{N290I}$ mutants, respectively, compared to the wild-type or complemented strains. Consistent with these findings, analysis of the transcription profile of the *cab1*$^{G351S}$, *cab1*$^{S158A}$, and *cab1*$^{N290I}$ mutants showed a significant downregulation of the genes involved in sulfur assimilation and the cysteine/methionine biosynthetic pathway compared to the wild type and mutant strains carrying a wild type *CAB1* gene (Figs. 4d, e and S2 and S3). Among these genes, the expression of the ATP sulfurylase-encoding gene, *MET3*, α-subunit of sulfite reductase gene, *MET10*, cystathionine β-synthase gene, *CYS4*, bifunctional dehydrogenase and ferrochelatase gene, *MET8*, and sulfate permease gene, *SUL2*, in each of the mutants decreased dramatically (between 75% and ~95%) compared to the wild type strain (Figs. 4e and S2).

### Inhibition of fungal ACS2 or V-type ATPase enzymes increases susceptibility to antifungals

The genetic data described above demonstrated a direct role of pantothenate utilization and CoA biosynthesis in yeast susceptibility to antifungals through alteration of vacuolar detoxification. Considering the potential implication of these findings on fungal therapy and reversal of multidrug resistance, we assessed whether a pharmacological approach using compounds that inhibit Cab1 activity or downstream steps such as AcCoA synthesis or vacuolar V-ATPase could mimic these genetic findings and usher in a new antifungal treatment modality. Analysis of the transcription

profile of the *cab1* mutants showed a dramatic decrease (12.6%, 12.9%, and 44.6% of that of the wild type) in the expression of the *ACS1* gene encoding one of two yeast AcCoA synthetases (Fig. S3). Interestingly, a yeast mutant carrying the *ACS2* gene under the regulatory control of the tet-off promoter (*acs2*-tet-off) was highly susceptible to caspofungin, fluconazole and terbinafine following addition of doxycycline (Fig. 5b). The MIC$_{50}$ for caspofungin shifted from 16 ng/ml in the absence of doxycycline to 10 ng/ml in the presence of the compound; that for fluconazole from 4.8 μg/ml to 0.06 μg/ml; and that for terbinafine from 4.5 μg/ml to 0.005 μg/ml. Consistent with these data, the celecoxib derivative AR-12, which is also a potent inhibitor of fungal AcCoA synthetases[25,26] increased yeast susceptibility to caspofungin (MIC$_{50}$ shift from 16 ng/ml to 3 ng/ml in the absence vs presence of AR-12), fluconazole (MIC$_{50}$ shift from 14.6 μg/ml to 0.9 μg/ml), and terbinafine (MIC$_{50}$ shift from 3.6 μg/ml to 0.07 μg/ml) (Figs. 5c and S4). Finally, because of the major alteration in vacuolar function and morphology in mutants altered in Cab1 activity, we examined the susceptibility of wild-type *S. cerevisiae* to caspofungin, fluconazole, and terbinafine in the absence or presence of concanamycin A, a known inhibitor of the vacuole V-Type ATPase[27,28]. As shown in Fig. 5d, treatment of yeast cells with concanamycin-A potentiates their susceptibility to caspofungin (MIC$_{50}$ shift from 36 μg/ml to 6.5 ng/ml), fluconazole (MIC$_{50}$ shift from 8.1 μg/ml to 5.7 μg/ml), and terbinafine (MIC$_{50}$ shift from 3.9 μg/ml to 0.9 μg/ml).

### Small molecule modulation of the PCA pathway leads to increased susceptibility of pathogenic fungi to antifungal drugs

The genetic and pharmacological data described above suggest that inhibition of specific steps in the CoA biosynthesis pathway or downstream steps leading to the regulation of vacuolar detoxification could be a promising therapeutic strategy for the treatment of fungal infections to enhance the potency of approved drugs while reducing their toxicity. Therefore, we screened a library of known PanK and CoA biosynthesis modulators to search for compounds that could render pathogenic fungi susceptible to clinically approved antifungal drugs. The pantazine analog, PZ-2891, an orthosteric activator of human PanK3[17], showed the highest potentiation among all compounds tested. Unlike other known Cab1 inhibitors[12,16], PZ-2891 did not inhibit pantothenate utilization of both WT and *cab1* mutant enzymes (Fig. S5). Furthermore, the compound had no antifungal activity against *S. cerevisiae*, *C. albicans* or *A. fumigatus* at concentrations up to 50 μM (Figs. 6, S6 and S7d). Interestingly, combinations of PZ-2891 with either amphotericin B, caspofungin or terbinafine at sublethal concentrations resulted in dramatic increases in the susceptibility of *S. cerevisiae* and *C. albicans* to these drugs (Figs. 6a, b and S6a, b). Similarly, PZ-2891 was found to increase the susceptibility of *A. fumigatus* to caspofungin (Figs. 6c, d and S6c). Unlike its inhibitory activity of human PanK3 in the absence or low levels of AcCoA, our data showed that PZ-2891 had little to no effect on Cab1 activity in vitro in the absence or presence of AcCoA (Fig. S7). Instead, the steady state levels of CoA following treatment with the compound increased by ~1.7-fold (Fig. 7a). These findings suggest that the mechanism of drug potentiation mediated by PZ-2891 in yeast could be through inhibition of CoA utilization, potentially by blocking the conversion of CoA to AcCoA by Acs1, which is not essential for yeast viability on glucose-based media. Therefore, we examined the direct inhibition of Acs1 activity by PZ-2891 using a hydroxylamine-coupled assay[26,29]. As shown in Fig. 7b, Acs1 activity was inhibited by PZ-2891 with 32.7% inhibition of the

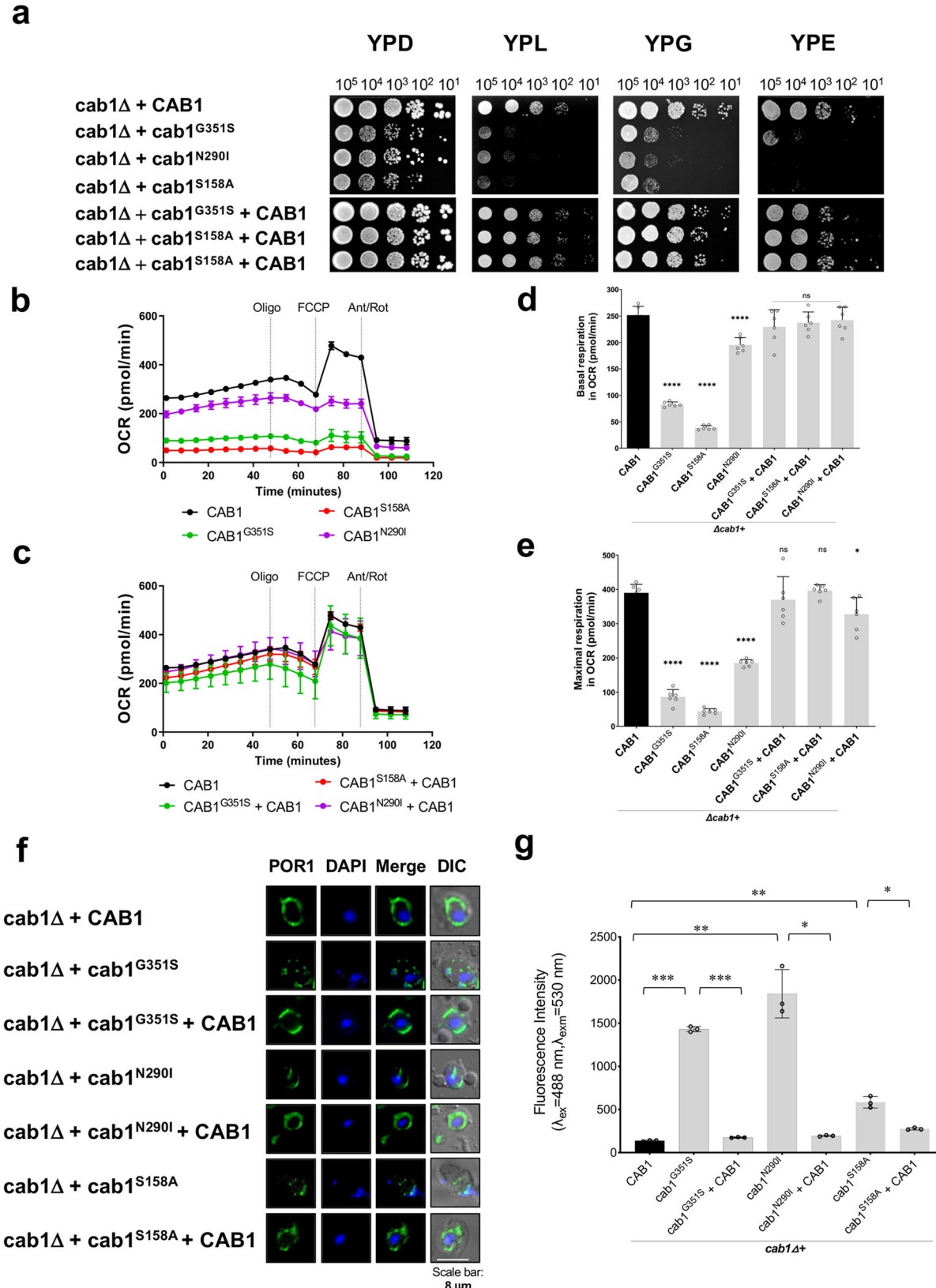

enzyme activity at 18.8 µM (Fig. 7b). As a control, and consistent with previous studies[26], the activity of purified yeast Acs1 in vitro was inhibited by AR-12 with a calculated $IC_{50}$ of ~18 µM (Fig. 7b). In silico molecular docking analysis of Acs1p with PZ-2891 indicates that PZ-2891 binds to the AMP binding pocket of the enzyme as shown in Fig. 7c–e. The binding of

AMP has been shown to induce a conformational change in the active site, facilitating the subsequent enzyme reaction[30,31]. Based on this analysis, PZ-2891 appears to compete with AMP, thus preventing Acs1 enzyme from undergoing conformational changes necessary for enzyme activity and catalysis. Together, these data demonstrate that the mechanism of drug

**Fig. 3 | Yeast *cab1Δ* strains harboring various *CAB1* mutations display defects in mitochondrial function. a** Solid growth assays reveal that mutations in *CAB1* alter strains' ability to utilize non-fermentable carbon sources. Solid growth assays using ten-fold serial dilutions plated onto YPD (glucose), YPL (lactate), YPG (glycerol), or YPE (ethanol) media and observed over 3-4 days. The representative images are from three independent experiments, each performed in duplicates. **b, c** Oxygen consumption rate (OCR) of yeast cells harboring *CAB1* variants. OCR profile of *cab1Δ* strains using Seahorse 96X and Mito Stress kit. Dashed lines represent the injections of mitochondrial uncoupling drugs; oligomycin, FCCP, antimycin A, and rotenone. **d, e** Basal respiration and maximal respiration of *cab1Δ* strains (t-tests

performed for each group comparing mutants to the wild type strain at $p = 0.05$). **f** *cab1Δ* strains exhibit mitochondrial structural defects. Immuno-fluorescence microscopy images of *cab1Δ* strains reveal aberrant mitochondria structures. The representative images are from two independent experiments, each performed in duplicates. **g** *cab1Δ* strains have increased levels of reactive oxygen species (ROS). ROS analysis was performed using dihydrorodamine 123. Each data plot represents the average of three biological samples with the corresponding standard deviation (±SD). (**d, e,** and **g**) Statistical significance was determined using t-test ($p = 0.05$) with GraphPad Prism and the corresponding $p$-values (***$p < 0.001$, **$p < 0.01$, *$p < 0.05$, and ns >0.05) are indicated.

potentiation of the pantazine analog PZ-2891 is through the alteration of a critical downstream step in the PCA pathway catalyzed by Acs1.

## Discussion

In this study, we demonstrate that the biosynthesis of CoA from pantothenic acid and the subsequent conversion of CoA to AcCoA (the PCA pathway) play a crucial role in the regulation of vacuolar homeostasis and xenobiotic detoxification. Consequently, inhibition of the PCA pathway confers increased susceptibility to antifungal drugs, thus revealing a therapeutic strategy for potentiation of frontline antifungal drugs to prevent fungal infections.

Examining the implications of altered PCA pathway is fundamentally important to the understanding of the biology of fungal pathogens, and to the best of our knowledge, this study unveiled previously unrecognized cell biological mechanisms regulated by this pathway. Our studies using three *CAB1* mutants, *cab1^G351S^*, *cab1^N290I^* and *cab1^S158A^*, demonstrated that genetic modulation of pantothenate phosphorylation results in enhanced susceptibility to xenobiotics including metals and commonly used antifungal drugs including both drugs that target ergosterol biosynthesis inhibitors (terbinafine, fluconazole and Amphotericin B) and unrelated pathways (caspofungin, hygromycin B, cycloheximide). Although the *cab1* mutants themselves exhibited a slight decrease in growth compared to the WT strain, their susceptibility to antifungal drugs was significantly higher. For instance, whereas *cab1Δ+cab1^G351S^* displayed a 20% reduction in cell growth at 30°C compared to the *cab1Δ + CAB1^WT^* strain, the mutant showed increased susceptibility to various antifungal drugs, with susceptibility being approximately 12-fold for fluconazole, 15-fold for terbinafine, 3-fold for amphotericin, 5-fold for caspofungin, 23-fold for hygromycin B, and 16-fold for cycloheximide, compared to the WT strain.

In fungi, broad-spectrum susceptibility to drugs and other xenobiotics occurs when drug detoxification mechanisms, such as those mediated by the vacuole are altered. Consistent with this observation, our studies demonstrated that mutations altering Cab1 activity also led to significant changes in vacuolar and mitochondrial biogenesis and morphology. These findings indicate that the PCA pathway plays a crucial role in regulating vacuolar biogenesis and drug detoxification. Further supporting these findings, the broad-spectrum susceptibility to antifungals caused by mutations in the *CAB1* gene could be replicated by downregulating genes and pharmacologically inhibiting enzymes downstream of the PCA pathway. In yeast, AcCoA can be formed from CoA through multiple routes, including by AcCoA synthetases Acs1 and Acs2 (See Fig. 1a). Cells lacking both *ACS1* and *ACS2* genes are inviable, as are cells lacking *ACS2* in glucose medium since *ACS1* is subject to glucose repression[32,33]. We found that downregulation of the *ACS2* gene, as well as inhibition of Acs1/2 activity by the inhibitor AR-12, results in increased susceptibility of yeast cells to fluconazole, terbinafine and caspofungin.

While the exact mechanism by which the PCA pathway regulates vacuole-mediated drug detoxification remains unclear, previous studies have identified 292 genes involved in either positive (35 genes) or negative (257 genes) interactions with a *cab1* thermosensitive mutant (*cab1-5001*)[34–36]. Among these, three genes involved in ergosterol biosynthesis were identified. *ERG3*[37], which encodes C-5 sterol desaturase, was found to engage in a positive interaction with *cab1*, while *ERG13*[38], which encodes

HMG-CoA synthase, the second step in ergosterol synthesis from AcCoA substrate, and *ERG11*[38], which encodes Lanosterol 14-alpha-demethylase, were found to engage in negative interactions with *cab1-5001*. Ergosterol biosynthesis is one of several metabolic pathways that rely on the production of AcCoA[8]. Previous reports have linked disruption of ergosterol synthesis to altered vacuolar ATP-powered H⁺ pumps (V-ATPase) and vacuolar acidification[39]. Ergosterol is suggested to directly modulate the activity of V-ATPase, though the molecular mechanism remains to be fully elucidated[39]. Interestingly, inhibition of the V-type ATPase by concanamycin A significantly increased fungal susceptibility to caspofungin and terbinafine but had only moderate effect on fluconazole susceptibility (Fig. 5d). These data suggest that the PCA pathway could regulate vacuolar function through inhibition of ergosterol biosynthesis.

Our studies also showed that the yeast *cab1^G351S^*, *cab1^N290I^* and *cab1^S158A^* mutants with reduced pantothenate kinase activity have altered vacuolar morphology. This finding aligns with previous reports documenting associations between vacuolar dysfunction and various changes in morphology, including enlarged and fragmented vacuoles[19,20,40–42]. Interestingly, analysis of the 257 genes involved in negative genetic interactions with a *cab1* mutant[35] identified several genes involved in autophagy (*ATG10, ARO2, ATG15, ATG5, ATG7, ATG16, ATG3, ARO7*, and *ARO1*), vesicular fusion (*VPS39, YPT7, YKT6,* and *VAM7*) and degradation of inner vesicles within the vacuole (*PEP4* and *ATG15*) (Fig. S8). The autophagy pathway has previously been shown to be activated under nutrient-deprived conditions, leading to the formation of autophagosomes[43,44], which ultimately fuse with the vacuole[45]. Based on available data, we propose that reduced CoA and, consequently AcCoA levels, resulting from alterations in the PCA pathway trigger a cascade of events leading to ergosterol deficiency, vacuolar dysfunction, and subsequent loss of vacuole drug detoxification ability (Fig. 8).

Our genetic analysis of the PCA pathway has been instrumental in the development of an antifungal strategy using potentiators to enhance the activity of clinically approved drugs. We found that the pantazine PZ-2891 potentiates the antifungal activity of amphotericin B, caspofungin and terbinafine in *S. cerevisiae, C. albicans* and *A. fumigatus* (Fig. 6a, b). This potentiation, which applies broadly to antifungals with varied mechanisms of action and to metals, suggests a broad-based disruption of fungal cells' ability to detoxify drugs. While PZ-2891 has been shown to be an orthosteric activator of human PanK3, our studies demonstrated that it has no effect on fungal Cab1 activity (Fig. S7b, c). Interestingly, we found that cellular CoA levels in *S. cerevisiae* increased significantly following treatment with PZ-2891 due to inhibition of Acs1 by the compound (Fig. 7a, b). The activity of PZ-2891 against Acs1 is similar to that of the Celecoxib derivative AR-12, which has previously been shown to improve fluconazole activity in a murine model of Cryptococcosis[25]. The Acs1 and Acs2 proteins are highly conserved among different fungal pathogens. For instance, the Acs1 protein of *S. cerevisiae* shares ~82% similarity with its counterpart in *C. albicans*; 81% with that of *C. auris*, and 76% with that of *A. fumigatus*. Similarly, the Acs2 protein in *S. cerevisiae* shares 83%, 83%, and 77% similarity with its counterparts in *C. albicans, C. auris,* and *A. fumigatus*, respectively. Clinical studies have found that an analog of PZ-2891, BBP-671 (NCT04836494), is largely safe with limited adverse events reported in humans[46,47]. Thus, this class

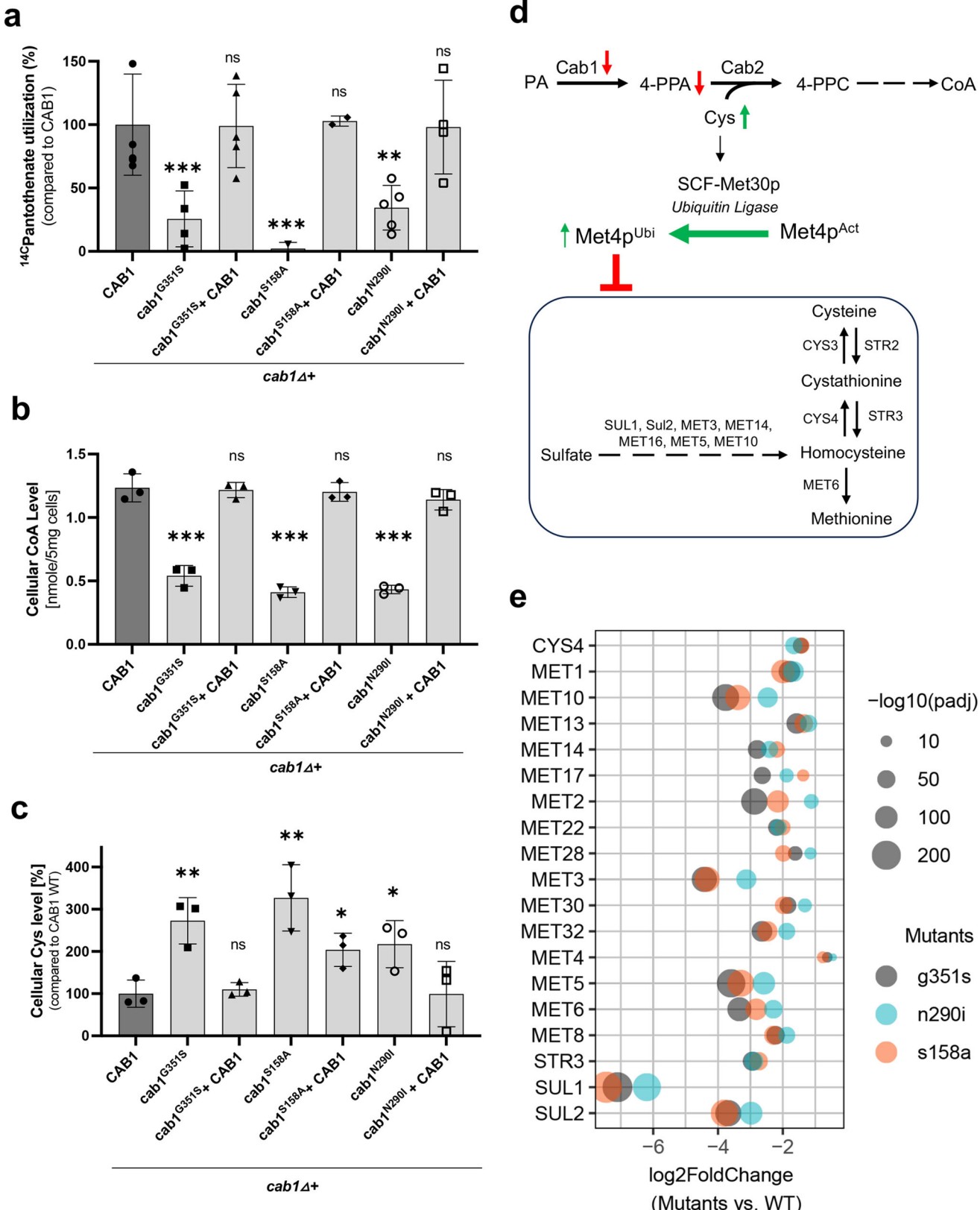

of small molecules (pantazines) may hold promise as the first antifungal adjuvants to enhance the potency of current drugs against drug-sensitive and -resistant strains while also lowering their toxicity.

In summary, our study shows that modulation of PanK activity results in impaired vacuolar homeostasis and xenobiotic detoxification, which in turns leads to enhanced fungal susceptibility to antifungal drugs. Therefore, compounds that target PanK or other key enzymes in the PCA pathway represent a promising path towards the development of novel therapeutic strategies to help combat the emerging threats posed by multi-drug resistant fungi and possibly other eukaryotic pathogens.

**Fig. 4 | Metabolic defects in yeast strains harboring *cab1* mutations. a** PanK activities in the *cab1Δ* strains harboring various *CAB1* mutations. Cell free extracts of yeast *cab1* mutants were used to measure the endogenous PA utilization of *cab1Δ + CAB1*, *cab1Δ + cab1^{G351S}*, *cab1Δ + cab1^{G351S} + CAB1*, *cab1Δ + cab1^{S158A}*, *cab1Δ + cab1^{S158A} + CAB1*, and *cab1Δ + cab1^{N290I}*, *cab1Δ + cab1^{N290I} + CAB1* strains using D-[1-^{14}C] pantothenate as a substrate for 10 min at 30 °C. Each data plot represents the average of three biological samples with the corresponding standard deviation (±SD). **b** Cellular levels of CoA in *cab1Δ* strains. CoA levels were measured using metabolite extracts from the yeast strains mentioned above grown in the presence of 0.2 μM PA. Each data plot represents the average of three biological samples with the corresponding standard deviation (±SD). **c** Cysteine cellular levels of *cab1Δ* strains harboring various *CAB1* mutations. Cellular cysteine levels were measured using the metabolite extracts from the yeast strain mentioned above grown in the presence of 0.2 μM PA. For these assays, t-test was done per each group compared to the parent *CAB1* wild-type strain (*p = 0.05*). Each data plot represents

the average of three biological samples with the corresponding standard deviation (±SD). **d** Schematic of the connection between the PCA pathway and the SUL/MET pathways. Transcription of the *SUL/MET* genes, responsible for synthesizing the crucial sulfur-containing amino acids methionine and cysteine, is mediated by the transcription factor Met4. This regulatory process is sensitive to changes in cellular cysteine levels. When cysteine levels increase, Met4 undergoes ubiquitination by the ubiquitin ligase Met30, leading to Met4's inactivation and subsequent repression of the *SUL/MET* genes. **e** RNA-Seq analysis for cysteine and sulfur homeostasis expressed in *cab1Δ* strains. Expression values are TMM normalized and adjusted *p*-values (padj) were calculated using the Benjamini-Hochberg method. The data represent the average of three biological samples. The illustrated volcano plot depicts log2-fold change values for the mutants, *cab1Δ + cab1^{G351S}*, *cab1Δ + cab1^{N290I}*, and *cab1Δ + cab1^{S158A}* in gray, cyan, and pink, respectively, against *cab1Δ + CAB1* WT, plotted against the negative logarithm of the padj values. The gene list with annotations shown in Table S1.

## Materials and methods

### Yeast strains and vectors
Yeast strains used in this study are shown in Table 1.

### Selection of yeast strains carrying *cab1* mutations
*cab1Δ*/pFL38-*CAB1* wild type and mutant strains were generated using plasmid shuffling as previously described from the parent strain *cab1Δ*/pFL39-*cab1^{G351S}* [16]. Briefly, pFL38-*CAB1* plasmids with wild type or mutant *cab1* (*cab1^{G351S}*, *cab1^{N290I}*, and *cab1^{S158A}*) were transformed into a *cab1Δ*/pFL39-*cab1^{G351S}* strain. Transformants were selected on minimal medium lacking uracil and tryptophan. Plasmid loss of pFL39-*cab1^{G351S}* from the strains were conducted by growing the strains in the minimal media supplemented with tryptophan and 5-fluoroanthranilic acid (5-FAA) but lacking uracil. The loss of the pFL39-*cab1^{G351S}* plasmid was confirmed by growth tests on medium lacking tryptophan. Add-back strains were generated by introducing pFL39-*CAB1* vector into yeast recipient strains.

### Growth assays
Yeast strains (WT and *cab1* mutants) were grown overnight at 30 °C in YPD medium and harvested (700 × *g* for 5 min at 4 °C), washed with water, and resuspended in 0.9% NaCl solution at OD_{600} of 0.5. Serial 10-fold dilutions were made and 5 μL of cell suspensions were spotted on YPD agar plates containing various antifungals (amphotericin B, caspofungin, fluconazole, terbinafine, hygromycin B and cycloheximide). For the respiratory growth assay, YP medium supplemented with ethanol, lactic acid or glycerol were used. Plates were incubated at 30 °C and the growth was monitored by image scan using the device ChemiDoc MP (Bio-Rad) every 24 h. For the liquid growth assay, the yeast strains were pre-grown as above and then diluted into 3 mL of yeast rich media supplemented with either 2% glucose (YPD), 2% glycerol (YPG) or 2% lactate (YPL) liquid media at the concentrations of 10 cells per μL and incubated at 30 °C by shaking at 230 rpm and the cell growth was monitored by optical density (OD_{600}). *A. fumigatus* growth was examined on GMM (1% glucose, 6 g/L NaNO_3, 0.52 g/L KCl, 0.52 g/L MgSO_4·7H_2O, 1.52 g/L KH_2PO_4 monobasic, 2.2 mg/L ZnSO_4·7H_2O, 1.1 mg/L H_3BO_3, 0.5 mg/L MnCl_2·4H_2O, 0.5 mg/L FeSO_4·7H2O, 0.16 mg/L CoCl_2·5H_2O, 0.16 mg/L CuSO_4·5H_2O, 0.11 mg/L (NH4)_6Mo_7O_{24}·4H_2O, and 5 mg/L Na4EDTA; pH 6.5).

### Electron microscopy analysis EM
Yeast strains (WT and *cab1* mutants) were grown overnight at 30 °C in YPD medium, harvested, and refreshed in YP media with 2% glycerol until reached OD_{600} of 1. The cells were harvested, washed, and used for high pressure freezing and freeze substitution for electron microscopy analysis. Unfixed samples were high-pressure frozen using a Leica HMP100 at 2000 psi. The frozen samples were then freeze substituted using a Leica Freeze AFS unit starting at −95 °C using 0.1% uranyl acetate in acetone for 50 h to −60 °C, then rinsed in 100% acetone and infiltrated over 24 h to −45 °C with Lowicryl HM20 resin (Electron Microscopy Science). Samples were placed in gelatin capsules and UV hardened at −45 °C for 48 h. The blocks

were allowed to cure for a further few days before trimmed and cut using a Leica UltraCut UC7. The 60 nm sections were collected on formvar/carbon-coated nickel grids and contrast stained using 2% uranyl acetate and lead citrate. The 60 nm sections on grids were viewed FEI Tecnai Biotwin TEM at 80 kV. Images were taken using AMT NanoSprint15 MK2 sCMOS camera.

### Pantothenate kinase (PanK) assay using ^{14}C-labeled PA
Pantothenate kinase assay using labeled PA was performed as previously described[13,14]. Briefly, cell-free extracts from yeast expressing Cab1 variants were obtained by homogenization, followed by centrifugation at 700 × *g* for 5 min. The 40 μL enzyme reaction contained reaction buffer (100 mM Tris HCl, 2.5 mM MgCl_2, 2.5 mM ATP, pH 7.4), D-[1-^{14}C] pantothenate (2 nmol, 0.1 μCi), and 144 μg cell-free extracts. The lysates total protein content was determined using the Bradford assay. The reaction was done at 30 °C for 10 min following the addition of 4 μL of 10% acetic acid to stop the reaction. The reaction mixture was spotted on a DE-81 filter (0.6 mm in diameter) placed within a spin column with a 2 mL collection tube. Following 5 min incubation, the spotted filters were centrifuged for 20 s at 700 × *g*, washed twice with 1% acetic acid in ethanol, and collected for liquid scintillation spectrometry.

### Cellular CoA determination
The determination of cellular CoA levels in yeast strains expressing different Cab1 variants was done using soluble metabolites extractions from *S. cerevisiae* as previously described[48]. Briefly, the yeast strains were inoculated in 3 mLs of YPD liquid media and grown overnight at 30 °C in a shaking incubator. Cells were harvested by centrifugation at 2000 × *g* for 5 min, resuspended in fresh media, and diluted to an OD of 0.2 in 10 mL YPD media. The cultures were grown until an OD of 0.8–1.0 was reached. Cells were harvested by centrifugation at 2000 × *g* for 5 min, washed twice with 60% methanol. Cell pellets were resuspended in 1 mL of 75% ethanol, extracted for 3 min at 85 °C with intermittent vortexing, then cooled rapidly without freezing. The ethanolic extract was separated from the cell debris by centrifugation at 4000 × *g* for 5 min, and the supernatants were evaporated to dryness in a vacuum centrifuge. The dried metabolites were resuspended in water (0.5 mL per 0.1 g cell weight), and insoluble particles were removed by centrifugation at 4000 × *g* at 4 °C for 10 min. The aqueous extract was stored at −80 °C. Metabolites extracts were then used in Coenzyme A detection kit (Sigma) to quantify cellular CoA.

### Cellular cysteine determination
The determination of cellular cysteine levels in yeast strain producing different Cab1 variants was done using soluble metabolites extractions from *S. cerevisiae* as described above. Metabolites extracts were then used in fluorometric cysteine assay kit (Abcam).

### RNA sequencing and data analysis
RNA samples from yeast strain expressing different Cab1 variants were extracted using YeaStar RNA kit (Zymo Research). The RNA samples

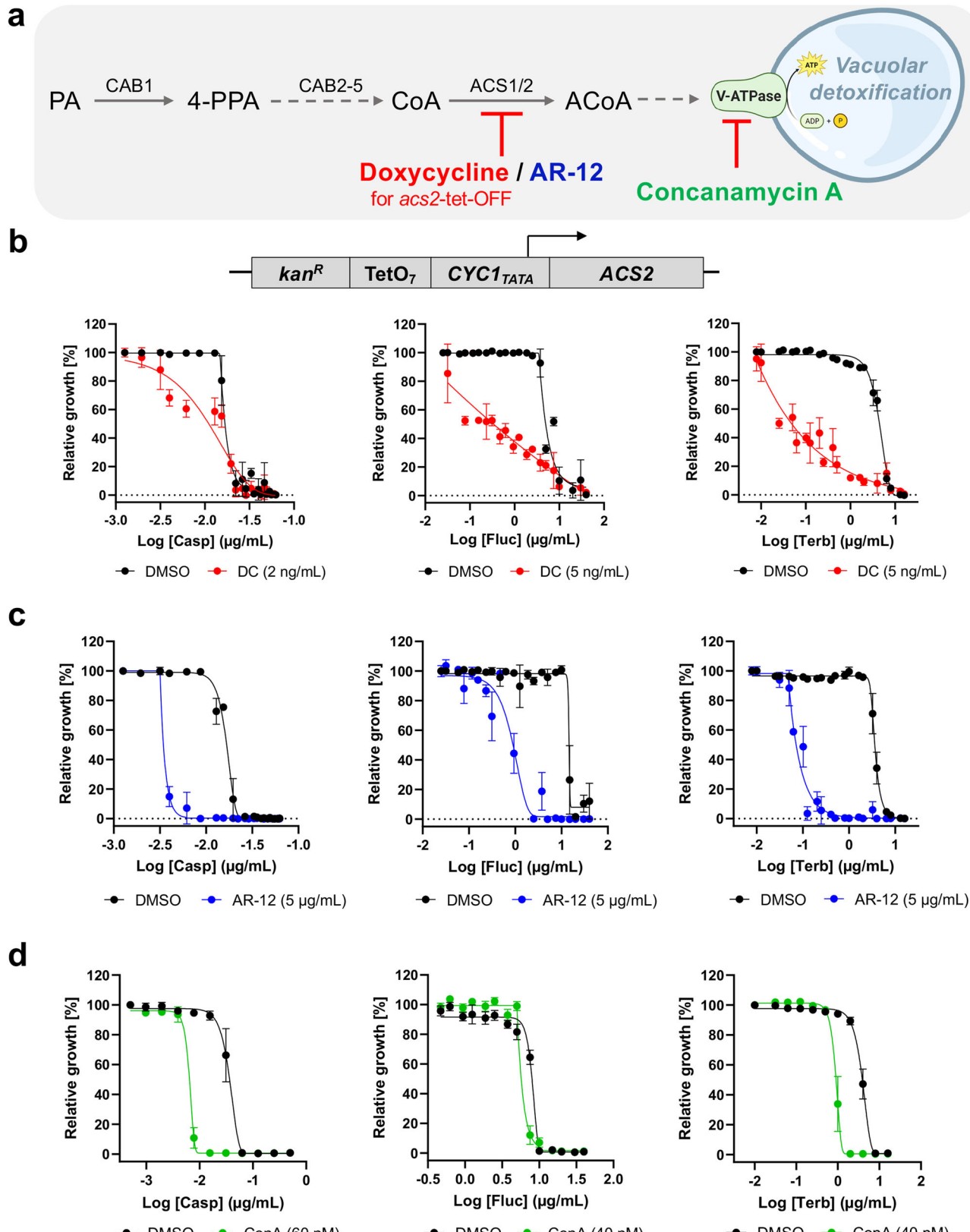

**Fig. 5 | Chemical inhibition of Acs2 and V-type ATPase increases susceptibility to a variety of AFDs. a** Schematic of the relevant portions of the PCA pathway. The vacuolar structure was created using BioRender (BioRender.com). **b** The growth of *acs2* tet-off strain is inhibited by increasing concentrations of doxycycline. *S. cerevisiae acs2* tet-off strain was inoculated in the presence or absence of doxycycline and normalized to DMSO-treated wells (no drug=100% growth) and 200 μM amorolfine (0% growth). **c** AR-12 potentiates caspofungin, fluconazole, and terbinafine by

factors of ~100x against *S. cerevisiae* WT. **d** Concanamycin A increases yeast susceptibility to antifungal drugs. *S. cerevisiae* WT was inoculated in the presence or absence of concanamycin A at 30 °C for 48 h. The growth was normalized to DMSO-treated wells (no drug = 100% growth) and 200 μM amorolfine well (0% growth). Liquid growth assays were conducted in quadruplicate (n = 4) and the plotted graphs represent the average of 4 data sets ± SD.

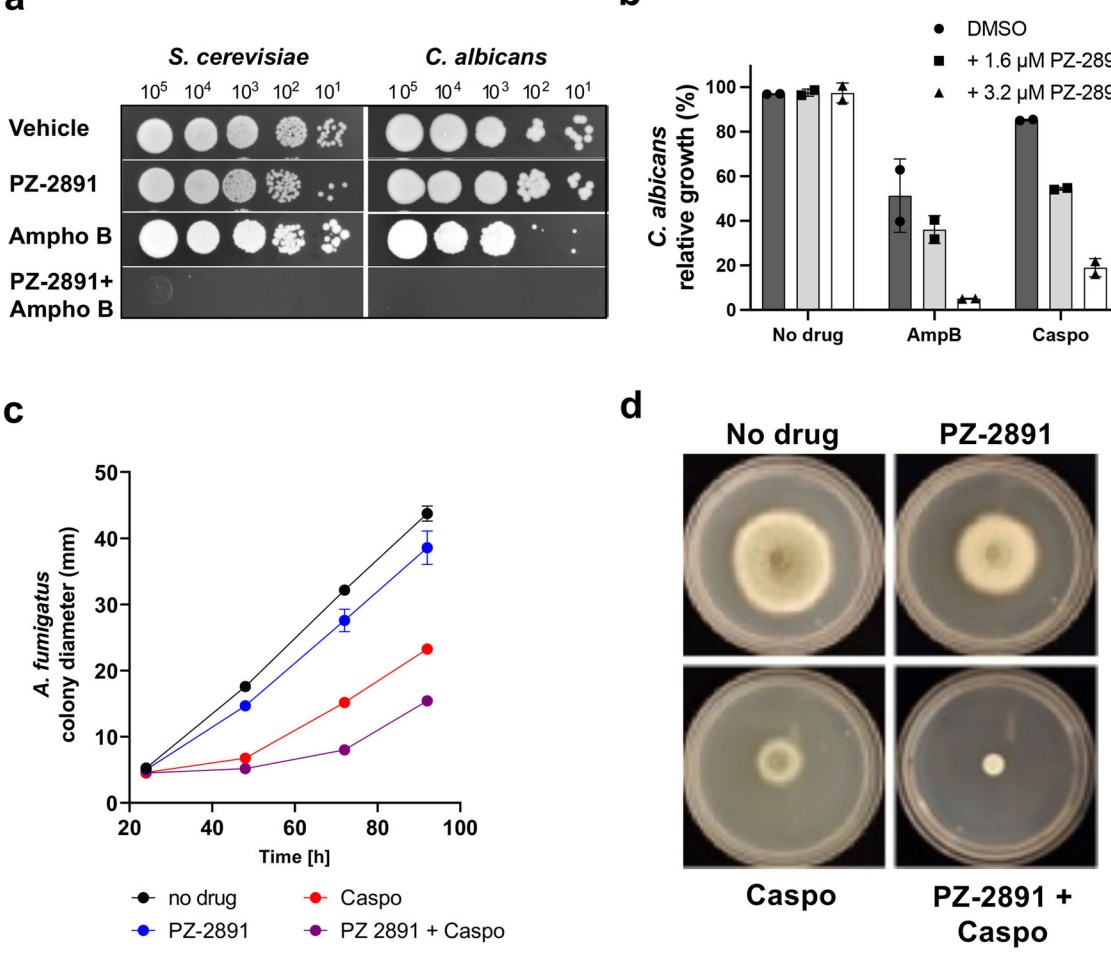

**Fig. 6 | PZ-2891 increases AFD susceptibility in *S. cerevisiae*, *C. albicans* and *A. fumigatus*. a** Solid growth assays showing increased *S. cerevisiae* and *C. albicans* susceptibility to amphotericin B when potentiated by PZ-2891. The representative images are from two independent experiments, each performed in duplicates. **b** Liquid growth assays showing increased *C. albicans* susceptibility to amphotericin B (125 ng/mL) and caspofungin (3.9 ng/mL) is potentiated by PZ-2891. The growth was normalized to DMSO-treated wells (no drug=100% growth) and 200 μM amorolfine well (0% growth). Liquid growth assays were conducted in quadruplicate (n = 4) and the plotted graphs represent the average of 4 data sets ± SD. **c, d** Average colony diameter rate and captures of solid media assay (up to 72 h of growth) with *A. fumigatus* cells under caspofungin treatment (20 μg/ml) in the presence or absence of PZ-2891 (50 μM). The data in Fig. 6c represent an average of 3 independent experiments ± SD. The representative images in (d) are from three independent experiments.

utilized in this study comprised three biological replicates from each of seven different classes of yeast strains. These classes include: 1) *cab1Δ* + *CAB1*, 2) *cab1Δ* + *cab1^G351S^*, 3) *cab1Δ* + *cab1^G351S^* + *CAB1*, 4) *cab1Δ* + *cab1^S158A^*, 5) *cab1Δ* + *cab1^S158A^* + *CAB1*, 6) *cab1Δ* + *cab1^N290I^*, and 7) *cab1Δ* + *cab1^N290I^* + *CAB1*. RNA sequencing was conducted by Yale Center for Genome Analysis (YCGA). RNA quality and integrity was determined by nanodrop and by resolving an aliquot of the extracted RNA on Agilent Bioanalyzer gel, respectively. RNA integrity number (RIN) values of the analyzed samples ranged from 8.9 to 10, exceeding the minimum required value of 7 for library preparation. For cDNA library preparation, the mRNAs were isolated from approximately 200 ng of total RNA using KAPA mRNA HyperPrep Kit (Roche Molecular Systems, Inc). Following first-strand synthesis with random primers, second strand synthesis and A-tailing were performed with dUTP to generate strand-specific sequencing libraries. Finally, library amplification amplified fragments carrying the appropriate adapter sequences at both ends. Indexed libraries that met appropriate cut-offs for both were quantified by Kapa Biosystems qRT-PCR reagents and kits (Millipore Sigma). Samples were sequenced using 100 bp paired-end sequencing on an Illumina NovaSeq according to Illumina protocols. FastQC (0.11.9, llumina, Inc) was used to check the quality of the raw reads. TrimGalore (0.6.7, Babraham Bioinformatics) was used for the removal of low-quality reads (reads with Quality Phred score <20 or reads shorter than 20 bp in length) and adapter sequences. The filtered and trimmed paired-end reads were aligned to the reference genome of *Saccharomyces cerevisiae* (R64-1-1) using HISAT2 (2.2.1)[49]. The number of reads in the Bam files that overlap with gene features were counted using the featureCounts function in the Subread package (2.16.0). Differential gene expression was performed using DEseq2 in Bioconductor version: Release (3.18). Counts were converted into counts per million using the cpm function in edgeR in Bioconductor 3.18 release. The Bubble plot in Fig. 4e and the heatmap in Fig. S2 were produced using ggplot2 (3.5.0) and pheatmap (1.0.12), respectively, with Rstudio (2023.12.1) and R (4.3.3).

**Respirometry analysis**

The oxygen consumption rate (OCR) of yeast strains expressing different Cab1 variants was determined using Seahorse 96X and Mito Stress kit. Yeast strains (WT and *cab1* mutants) were grown overnight at 30 °C in YPD medium, harvested, and refreshed in SC medium supplemented with 2% glucose until reached OD_{600} of 0.6. Then, cells were harvested, washed, and seeded ($6 \times 10^4$ cells per well) in Seahorse XFp plates coated with poly-Lysine (50 μL of 0.1 mg/mL). A minimum of 8 technical replicates were performed for each experiment at 30 °C. The seeded plate was centrifuged at 500 rpm for 5 min to promote yeast adhesion and the plate was rested for 30 min at RT. A soaked and calibrated Seahorse XF96 Sensor Cartridge was

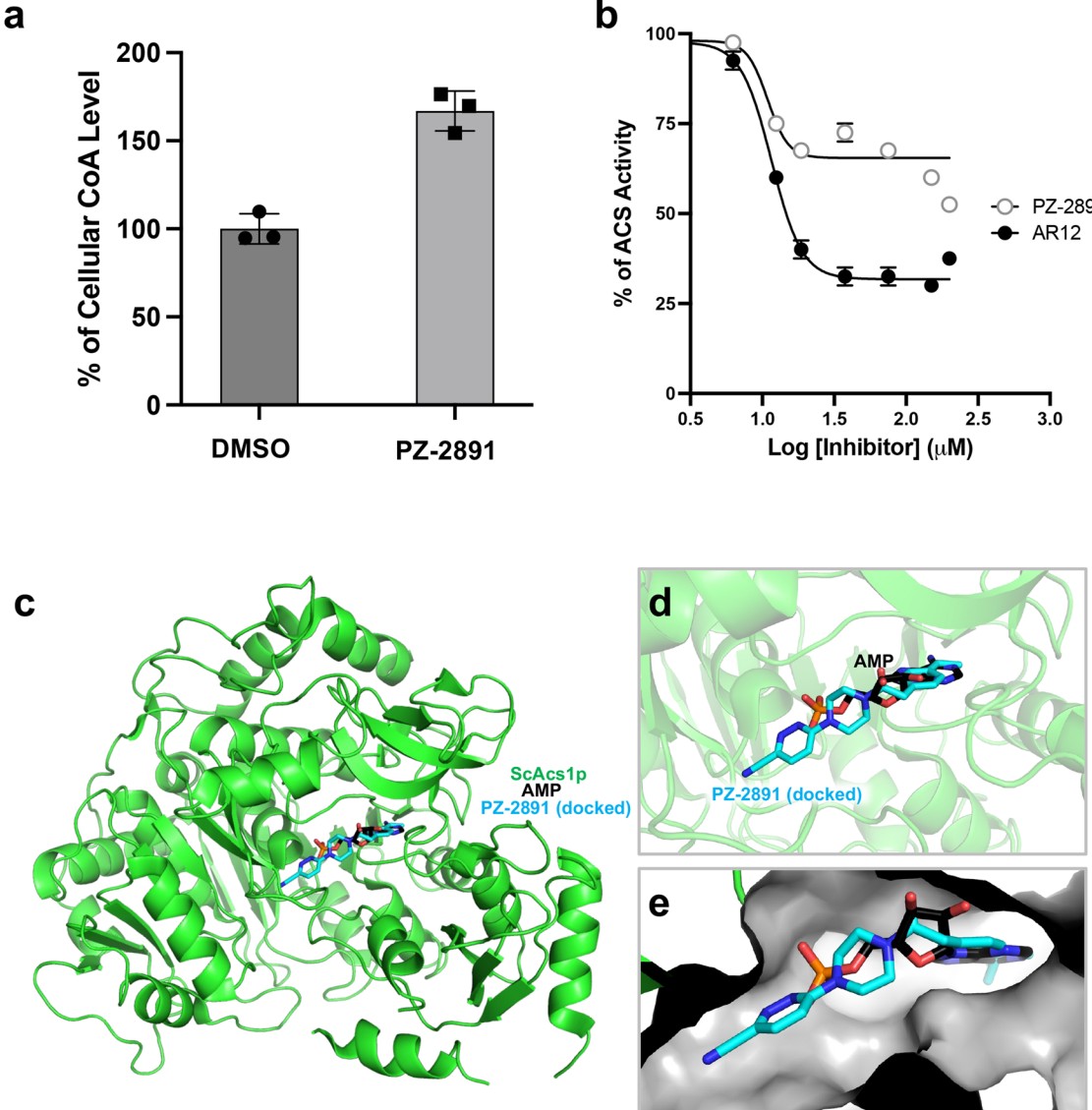

**Fig. 7 | PZ-2891 inhibits acetyl CoA synthetase activity. a** Cellular CoA levels in *S. cerevisiae* following treatment with PZ-2891. CoA levels were measured using the metabolite extracts from the *S cerevisiae* cells grown in minimal glucose medium supplemented with 1 μM PA in the presence or absence of 50 μM PZ-2891. Each data plot represents the average of three biological samples with the corresponding standard deviation (±SD). Statistical significance was determined using t-test (*p = 0.05*) with GraphPad Prism and the corresponding *p* values (**p < 0.01) are indicated. **b** Yeast acetyl CoA synthetase activity is inhibited by PZ-2891. The in vitro activity of purified enzyme from *S. cerevisiae* was measured using a standard hydroxylamine-coupled assay, at 37 °C for 30 min, in the presence of various concentrations of AR-12 or PZ-2891. The activities in the presence of inhibitors are expressed as a percentage of activity compared to the 100% activity observed in the DMSO mock control. The data represents the average of three independent experiments (±SD). **c–e** Binding prediction of PZ-2891 to ScAcs1p structure in silico and the inhibition kinetics. **c** The apo form of ScAcs1p structure (PDB:1RY2) was used for molecular docking with PZ-2891 using AutoDock Vina. ScAcs1p chain is represented in green cartoon, docked PZ-2891 is presented in cyan sticks, and the AMP solved bound to ScAcs1p (PDB:1RY2) is presented in black sticks. **d** Represents the binding of PZ-2891 in the AMP binding pocket of ScAcs1p. **e** Shows the configuration of PZ-2891 and AMP in the binding site of ScAcs1p (gray surface).

prepared before loading into the Seahorse XF96 analyzer (Agilent) which determined the cells basal OCR and following the injection of mitochondrial uncoupling drugs; oligomycin (5 μM), carbonyl cyanide-4 (tri-fluoromethoxy) phenylhydrazone (FCCP) (10 μM), antimycin A (10 μM), and rotenone (5 μM). The readouts were normalized using nuclear Hoechst staining for the immobilized yeast cells.

### Yeast growth in the presence of antifungal drugs, inhibitors, and potentiators

To investigate the effect of common AFDs (amphotericin B, caspofungin, fluconazole, and terbinafine) in combination with compounds and potentiators (concanamycin A, doxycycline, PZ-2891, α-PanAm, AR-12) the yeast growth was monitored using a liquid assay in a 96-well plate. Overnight yeast precultures (WT strains or acs-Tet-Off when mentioned) were prepared in YPD medium at 30 °C. Cells were washed and refreshed in YPD until reaching $OD_{600}$ of 0.6. In a 96-well plate, cells ($10^3$ cells/mL, 100 μL final volume) were treated with decreasing concentrations (two-fold dilutions) of AFDs, and different dosages of compounds and potentiators. For reference, amorolfine (200 μM) and DMSO (0.6%) were used as positive and negative controls to determine 100% and 0% growth inhibition, respectively. Plates were incubated at 30 °C. Optical density measurements were taken using a BioTek SynergyMx microplate reader every 12 h. Data are shown as mean ± SD of four independent experiments. Growth curves where visualized and determined from a sigmoidal dose-response curve using GraphPad Prism version 9.5.1 (GraphPad Software, San Diego, CA). Statistical significance was determined using t-test (*p = 0.05*) with GraphPad Prism.

## a

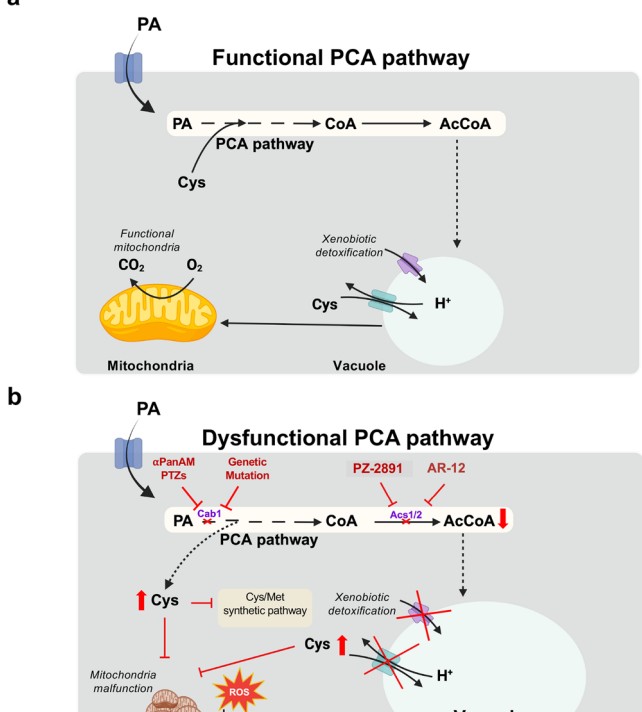

**Fig. 8 | Model for PCA pathway-mediated regulation of vacuolar detoxification and susceptibility to antifungal drugs in fungi.** We propose that the pathway for biosynthesis of CoA and AcCOA (the PCA pathway) from pantothenate has major regulatory function in the control of organellar biogenesis in fungi. **a** The upper panel illustrates the cellular events associated with the functioning PCA pathway. AcCoA and ergosterol are produced through the normal PCA pathway, with ergosterol supporting normal vacuole and mitochondrial function. **b** The lower panel illustrates the dysfunctional PCA pathway, resulting in severe downstream effects on vacuolar function. Vacuolar homeostasis and xenobiotic detoxification heavily rely on the functionality of the PCA pathway in fungal strains. A reduction in ergosterol synthesis inhibits the function of vacuole and results in defects in cysteine sequestration and impairments in drug/heavy metal detoxification. These vacuolar function defects also contribute to increased ROS levels and mitochondrial abnormalities. The pivotal role of the PCA pathway in these downstream cellular events creates a unique vulnerability in yeast strains that can be targeted through the use of potentiators (PAMS) like PZ-2891, in addition to other modulators of PanK, Acs, and V-ATPase enzyme activities. The figure was created using BioRender (BioRender.com) and PowerPoint (Microsoft.com). PA pantothenic acid, PTZ pyrimidone triazol, αPanAM alpha-methyl-N-phenethyl-pantothenamide, CoA Co-enzyme A, ROS reactive oxygen species, PAMS potentiators of antimicrobial susceptibility.

## Reactive oxygen species (ROS) content

Reactive oxygen species (ROS) in the *cab1* mutants were determined by change of oxidative status of fluorescence dye caused by ROS inside of the cell. ROS oxidize dihydrorhodamine 123 (DHR123; Sigma-Aldrich®, Darmstadt, Germany), which in turn produces green, fluorescent R123. To monitor ROS, cells were pre-grown overnight at 30 °C in YPD to the $OD_{600}$ of 0.5–1.0 and the cells were diluted to the OD of 0.4 and loaded with 1.25 µg/mL of DHR123 for 2 h at 30 °C. At the end of the incubation time, cells were harvested (2 min at $9000 \times g$) and re-suspended in water at the $OD_{600}$ of 0.05, and the fluorescence was quantified by a plate reader. For each sample, 100 µL of cell suspension was added into each well and the fluorescence was measured (excitation/emission spectra of 488/530 nm). Emission values from the control cells untreated with the dye were used as background for each strain. ROS generation for each *cab1* strains was measured as the percentage of fluorescence emission obtained from the *cab1Δ* strain harboring WT *CAB1* gene.

## PanK activity of recombinant Cab1 in the presence of PZ-2891 and AcCoA

His-tagged Cab1 recombinant enzyme was produced and purified as was previously described[16]. A Kinase-Glo (Promega) assay kit for kinase activity was used to determine the activity of the purified PanK under different conditions[16].

## Vacuolar visualization and cell size determination

To determine the ratio of vacuolar area over cell area, different yeast strains were stained with CellTracker™ Blue CMAC as explained in the methods above. Images were captured using fluorescence microscope and analyzed using Image J software. The cell surface area (in square pixel) and vacuolar surface area (in square pixel) were calculated in Image J and percentage of vacuolar area/cell area was calculated. A total of 100 cells were analyzed from each yeast strain. The data was plotted and analyzed in GraphPad Prism version 9.5.1 (GraphPad Software, San Diego, CA). Statistical significance was determined using Welch's t-test with GraphPad Prism.

## Radial growth assay and AFD sensitivity assays with *A. fumigatus*

The radial growth measurements of *A. fumigatus* were performed as previously described[16,50]. Briefly, 2 µL of a $2.5 \times 10^6$ mL$^{-1}$ conidial suspension of wild-type CEA10 *A. fumigatus* was point inoculated onto the center of a solid GMM in the absence or presence of 50 µM PZ-2891, 20 µg/mL caspofungin, and their combination. Plates were incubated for 96 h at 35 °C, with colony diameters measured and photographs taken each day.

## Acetyl CoA synthetase (ACS) activity assay

The ACS assay was performed by monitoring the formation of the adenyl acetate, the intermediate of the enzyme reaction, utilizing *S. cerevisiae* acetyl CoA synthetase (Sigma, A1765), following established protocols with some modifications[26,29]. In a 100 µL reaction volume, composed of 100 mM potassium phosphate at pH7.5, 5 mM $MgCl_2$, 2 mM ATP, 50 mM potassium fluoride, 10 mM reduced glutathione, 0.35 mM CoA, 10 mM potassium acetate, 200 mM neutralized hydroxylamine adjusted to pH7.3, 0.005 units of the enzyme, and the inhibitors (in 1% DMSO), the components were combined. The mixture was then incubated for 30 min at 37 °C. Termination of the reaction was achieved by addition of 50 µL of a solution containing ferric chloride (12 M) and trichloroacetic acid (12%). The resultant product, acethydroxamic acid, was quantified using a BioTek SynergyMx microplate reader at $OD_{540}$. The background correction was performed by utilizing a blank reaction comprising all the reaction components, which was subsequently terminated using acidified ferric chloride solution, without undergoing any incubation time.

## Genetic interaction analysis of *CAB1*

The list of 292 genes displaying genetic interactions with a *cab1* mutant was obtained from previous reports[34–36]. The gene IDs were converted into their corresponding Ensembl gene IDs using the conversion tool available at YeastMine (https://bluegenes.yeastgenome.org/yeastmine/upload/input). Enrichment analysis was performed using the ShinyGO tool (http://bioinformatics.sdstate.edu/go/), applying standard settings including an FDR cutoff of 0.05 and a minimum pathway size of two genes. Redundant pathways were removed to enhance clarity and relevance.

## Statistics and reproducibility

The number of biological samples utilized in the experiments was 3 and the corresponding *p*-values were provided in each figure legend. Data are presented as means ± standard deviation (SD). Statistical analysis was conducted using GraphPad Prism 9 (Graphpad software, CA, US). For

**Table 1 | Yeast strain and their vectors used in this study**

| Species | Strain | Genotype | References |
|---|---|---|---|
| *S. cerevisiae* | W303-1B | *Matα ade2-1 leu2-3,112 ura3-1 trp1-1 his3-11,15 can1-100 cab1Δ::KanMX4 + pFL38(URA3)-CAB1* | 15 |
| *S. cerevisiae* | W303-1B | *Matα ade2-1 leu2-3,112 ura3-1 trp1-1 his3-11,15 can1-100 cab1Δ::KanMX4 + pFL38(URA3)-cab1$^{G351S}$* | 16 |
| *S. cerevisiae* | W303-1B | *Matα ade2-1 leu2-3,112 ura3-1 trp1-1 his3-11,15 can1-100 cab1Δ::KanMX4 + pFL38(URA3)-cab1$^{G351S}$ + pFL39(TRP1)-CAB1* | This study |
| *S. cerevisiae* | W303-1B | *Matα ade2-1 leu2-3,112 ura3-1 trp1-1 his3-11,15 can1-100 cab1Δ::KanMX4 + pFL39(TRP1)-cab1$^{N290I}$* | 15 |
| *S. cerevisiae* | W303-1B | *Matα ade2-1 leu2-3,112 ura3-1 trp1-1 his3-11,15 can1-100 cab1Δ::KanMX4 + pFL39(TRP1)-cab1$^{N290I}$ + pFL38(URA3)-CAB1* | This study |
| *S. cerevisiae* | W303-1B | *Matα ade2-1 leu2-3,112 ura3-1 trp1-1 his3-11,15 can1-100 cab1Δ::KanMX4 + pFL38(URA3)-cab1$^{S158A}$* | 16 |
| *S. cerevisiae* | W303-1B | *Matα ade2-1 leu2-3,112 ura3-1 trp1-1 his3-11,15 can1-100 cab1Δ::KanMX4 + pFL38(URA3)-cab1$^{S158A}$ + pFL39(TRP1)-CAB1* | This study |
| *S. cerevisiae* | W303-1B | *Matα ade2-1 leu2-3,112 ura3-1 trp1-1 his3-11,15 can1-100 cab1Δ::KanMX4 + pFL38(URA3)-CAB1$^{G351A}$* | 16 |
| *S. cerevisiae* | W303-1B | *cab1Δ::KanMX4 + pFL38- CAB1$^{G351A}$ + pFL39(TRP1)-CAB1* | This study |
| *S. cerevisiae* | W303-1B | *Matα ade2-1 leu2-3,112 ura3-1 trp1-1 his3-11,15 can1-100 cab1Δ::KanMX4 + pFL38(URA3)-cab1$^{Y220A}$* | 16 |
| *S. cerevisiae* | W303-1B | *Matα ade2-1 leu2-3,112 ura3-1 trp1-1 his3-11,15 can1-100 cab1Δ::KanMX4 + pFL38(URA3)-CAB1$^{Y220A}$ + pFL39(TRP1)-CAB1* | This study |
| *S. cerevisiae* | JS91.15-23 | WT (*MATα ura3 leu2 trp1 his3*) | 13 |
| *S. cerevisiae* | JS91.14-24 | *cab1$^{ts}$* (*MATa ura3 his3 cab1$^{ts}$*) | 13 |
| *S. cerevisiae* | JS91.14-24 | *cab1$^{G351S}$ + p5472-CAB1* | This study |
| *S. cerevisiae* | R1158 | *acs2-tet-off* | 51 |
| *C. albicans* | SC5314 | Wild-type | 52 |
| *C. glabrata* | CSH10 CAG1 | Wild-type | N/A (clinical isolate) |
| *C. parapsilosis* | CAP-ATCC 22019 | Wild-type | Source BEI Resources |
| *A. fumigatus* | CEA10 | Wild-type | 53 |

experiments involving two groups, we applied Student's *t* test while ANOVA was employed for multiple comparisons. Fluorescence images were processed using the FiJi/Image J suite.

### Reporting summary

Further information on research design is available in the Nature Portfolio Reporting Summary linked to this article.

### Data availability

RNAseq data have been deposited in the NCBI Sequence Read Archive (SRA) [Bio project: PRJNA1080618, (SAMN40174357: *cab1 + CAB1*, SAMN40174358: *cab1Δ + cab1$^{G351S}$*, SAMN40174359: *cab1Δ + cab1$^{G351S}$ + CAB1*, SAMN40174360: *cab1Δ + cab1$^{S158A}$*, SAMN40174361: *cab1Δ + cab1$^{S158A}$ + CAB1*, SAMN40174362: *cab1Δ + cab1$^{N290I}$*, SAMN40174363: *cab1Δ + cab1$^{N290I}$ + CAB1*]. All supporting data generated for the graphs and charts presented in the main figures are included in the supplementary data. All other data are available from the corresponding author upon reasonable request.

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

## Acknowledgements

This research was supported by an award to C.B.M. by the Blavatnik Family Foundation. CBM research is also supported by funds from NIH and

the Steven & Alexandra Cohen Foundation. We thank Dr. Paola Goffrini for providing the yeast strain (*S. cerevisiae W303-1B cab1Δ*/pFL39-*cab1*$^{G351S}$), Dr. Hans-Hoachim Schuller for providing the *cab1*$^{ts}$ mutant and its parental strain, Dr. Frederick Roth for providing the yeast efflux-deficient mutant and its parental strains, Dr. Joseph C. Gennaro for his assistance with the analysis of RNAseq data, and Dr. Isaline Renard for her early investigations into the potentiating activity of modulators of the PCA pathway. This project has also been funded in whole or in part with Federal funds from the National Institute of Allergy and Infectious Diseases, National Institutes of Health, Department of Health and Human Services, under Contract No. HHSN272201700059C.

## Author contributions

Conceptualization, J.Y.C., S.G., M.M., P.S., P.H., K.F., and C.B.M.; Methodology, J.Y.C., S.G., M.M., P.S., and E.M.A.; Formal Analysis, J.Y.C., S.G., M.M., P.S., P.V., E.M.A., X.S., and O.K.; Investigation, J.Y.C., S.G., M.M., P.S., P.H., E.M.A., K.F., and C.B.M.; Resources, J.Y.C., S.G., M.M., P.S., P.H., E.M.A., K.F., and C.B.M.; Writing – Original Draft, J.Y.C., S.G., P.S., P.V., P.H., K.F., and C.B.M.; Writing – Review & Editing, J.Y.C., S.G., P.S., P.V., P.H., K.F., and C.B.M.; Visualization, J.Y.C., S.G., M.M., P.S., E.M.A., O.K., K.F., and C.B.M.; Supervision, P.H., K.F., and C.B.M.; Project Administration, P.H., K.F., and C.B.M.; Funding Acquisition, K.F. and C.B.M.

## Competing interests

The authors declare the following competing interests. C.B.M. is the founder of Curatix, which focuses on the development of anti-infectives. J.C.Y. conducted this work while an Associate Research Scientist at Yale. He is currently a Scientific Director Curatix. All other authors declare that they have no conflict of interest with the content of this article. A patent application on the use of the PAMS strategy to potentiate antifungal drugs has been submitted by CBM.
