## [Peer Review File · Communications Biology]

Reviewers' comments:

Reviewer #1 (Remarks to the Author):

Choi et al. examine the potential of PCA pathway inhibition to potentiate antifungal efficacy in *S. cerevisiae*. Using point mutants in the Cab1 protein, they demonstrate that thermosensitive variants of Cab1 confer hypersensitivity to a broad range of antifungals. They posit that inhibition of CoA and Acetyl-CoA may therefore offer an opportunity to potentiate antifungal activity and test this hypothesis in two major fungal pathogens, *C. albicans* and *A. fumigatus*. Using *S. cerevisiae* as a model system, they dissect a potential mode of action for cell sensitivity that involves inhibition of vacuolar integrity and drug detoxification, with mitochondrial fragmentation and increased oxidative stress as byproducts of this. The data presented support this conclusion.

Overall, the work is interesting and novel, and offers a potential strategy for combination therapy in fungal infections that will be of broad interest to the fungal pathogen community.

Given the argument that inhibition of the PCA pathway is a good potential therapeutic strategy, it would be interesting to see how well conserved the pharmacological targets (e.g. Acs1) are across the fungal pathogens mentioned in the text (*C. auris*, *C. albicans*, *A. fumigatus*, etc).

I have significant concerns about the RNAseq analysis. There is no indication how many biological replicates were performed. No QC data are reported for either RNA quality or sequencing quality. The methods specified suggest the analysis was done using an automated pipeline by a non-expert user. I suggest the authors seek support from their core facility or an expert collaborator to ensure their analysis is accurate, that the standard QC parameters are validated, and that their reported methods accurately reflect the rigour of their experimental design. In addition, RNAseq data should be made publically available through a standard repository such as the NCBI SRA.

Specific comments:

Fungi is large and diverse kingdom. The authors should specify which organisms they investigate in the abstract.

Please note that, if performed as currently described in the methods, all t-tests should be corrected for multiple comparisons, including the RNAseq dataset. The authors should ensure they perform the appropriate tests for the data analysed throughout the document. Please also note that experiments should be performed in at least biological triplicate, not only technical replicate (see line 462), as is standard practice to control for biological sources of experimental variation.

For all figures, the number of biological and technical replicates performed should be indicated, and statistical significance reported in the figure legend. Details of statistical tests should be indicated in the methods section.

Where representative images are shown, the number of biological replicates performed should be

indicated in the figure legends that accompany the representative image.

Line 120-124: Figure 1C is not referred to. Perhaps an error in line 124?

Typo line 186- measuring following

Methods: Lines 410-437 look to be from a core facility instruction manual. This should be corrected to describe specifically the methods applied in this dataset, not generic datasets. Lines 405-409 appear to be instructions for RNA quality analysis. The authors should report whether their RNA quality was within these standards. In addition, the title of this section should be changed to RNA quality control, not RNAseq quality control. Sequencing qc is performed post analysis.

- Library prep methods again appear to be standard procedures rather than the specific pipeline used. This should be clarified.

- standard RNAseq QC should be performed and parameters should be provided as supplementary material.

Reviewer #2 (Remarks to the Author):

In this study, the authors characterise the fungal pantothenate kinase as a target for the potentiation of antifungal drugs. They build on previous knowledge to show that tampering with the activity of the enzyme affect mitochondrial and vacuolar functionality, and potentiate the action of various antifungal drugs.

The work is well presented, the data is clear and the experimental approaches are well done.

However, in my humble opinion, and despite the high number of experiments performed, I believe this work does not add much new knowledge to the field. In addition, I believe the conclusions claimed by the authors are not sufficiently supported by the results shown.

I explain my major concerns here:

1. To my understanding the point mutation *cab1G351S* causes a temperature dependent phenotype, but the yeast should grow normally at 30°C. In contrast, in Figs. 1E and 2A a defect can be observed in basal growth, despite all growth assays being done at 30°C.
2. The effects of *cab1G351S* and *cab1N290I* on mitochondria and iron homeostasis (and more) have already been published (doi: 10.3390/ijms22010293). The fact that repressing pantothenate kinase activity potentiates antifungal activity was already shown by this manuscript's same authors (doi: 10.1016/j.str.2022.09.001)
3. I do not think the provided data prove a causal link between Cab1 and the vacuole. Although it is of course true that vacuoles participate in detoxification, the increased susceptibility to 'xenobiotics' could be caused by other of the pleiotropic effects of tampering with CoA synthesis (like metabolic effects and/or higher ROS, which the authors show and was already known). Thus, the stated in lines 287-288 is not true. The authors should check if the accumulation of any of the stressors inside vacuoles decrease in the *cab1* mutants, to prove that its function is impaired. According to Fig. 2B, the integrity of the vacuoles is fine (in contrast to what the authors repeatedly state). The size expansion could be caused by multiple reasons, including defects in growth or

oxidative stress.

4. It was also shown that AR-12 potentiates antifungal action (DOI: 10.1128/AAC.01061-16).

5. The data with PZ-2891 is very confusing. It was described as an activator of pantothenate kinases (in line 257 it is said inhibitor?). The authors state that they do not see an effect on the Cab1 enzymatic activity, but it causes increased accumulation of CoA, which would agree with increased activity. Is the enzymatic assay capable of detecting increase activity? The authors could measure cysteine to detect if the levels are reduced, in line with an increased kinase activity

Other points that need clarification are:

- The authors should explain the effects of the point mutations on protein activity.
- Given the slight growth defect of the cab1 mutants in normal YPD medium compared with the wild-type, the effect of iron and copper seem quite minor. Clearly not 'dramatically reduced' (line 153)
- It is surprising that PZ-2891, a drug described to modulate human PANK and modelled to bind the fungal enzyme by these authors, have no effect on Cab1 and does on Acs1. Can the authors back the inhibitory action on Acs1 with another enzymatic assay? Given their expertise, can they dock the binding of PZ-2891 to the enzyme? Can they measure cellular Acetyl-CoA to see if it's decreased?

Reviewer #3 (Remarks to the Author):

The manuscript by Choi et al. investigates a connection between vitamin B5 metabolism, vacuolar homeostasis and detoxification of xenobiotics. The authors mainly use genetic methods to show that the PCS pathway is a good target to increase the activity of various xenobiotics. In addition, the authors identify the pantazine analog PZ-2891 as a valid inhibitor of the ACS in yeast. In summary, I think this is a carefully executed study that yields some very interesting findings for the target readership. In summary, I would recommend the manuscript for publication with minor additions.

Most of the genetic data are very well executed and largely convincing. However, most of the experiments yield indirect evidence.

Could the data be further supported by the analysis of high throughput genetic data from yeast. Multiple genetic interaction studies from yeast are available and are vast sources of data. Could the authors systematically validate some of the obtained results in these high throughput datasets. For example the lipid EMAP and others come to mind that should have the observed data.

Can some of the data be confirmed by measuring the output of the inhibited pathway. For example, could the authors measure ergosterol levels in terbinafine treated cells with and without the potentiating effects?

What is the difference between the Cab1 mutants. I think a better explanation of the three different mutations and their effects on protein function would be useful.

The in vitro Acs1 activity assay could be improved in my opinion. Could the authors provide some kinetic data using for example different amounts of substrates?

We thank the reviewers for their thorough review of our manuscript titled "Vitamin B5 Metabolism is Essential for Vacuolar and Mitochondrial Functions and Drug Detoxification in Fungi". Their suggestions and comments have guided this revision and strengthen the message and conclusions of the paper and helped improved the overall quality of the work.

Our point-by-point response (in blue) to the reviewers (in black) comments are as follows:

Reviewer 1,

Choi et al. examine the potential of PCA pathway inhibition to potentiate antifungal efficacy in *S. cerevisiae*. Using point mutants in the Cab1 protein, then demonstrate that thermosensitive variants of Cab1 confer hypersensitivity to a broad range of antifungals. They posit that inhibition of CoA and Acetyl-CoA may therefore offer an opportunity to potentiate antifungal activity and test this hypothesis in two major fungal pathogens, *C. albicans* and *A. fumigatus*. Using *S. cerevisiae* as a model system, they dissect a potential mode of action for cell sensitivity that involves inhibition of vacuolar integrity and drug detoxification, with mitochondrial fragmentation and increased oxidative stress as byproducts of this. The data presented support this conclusion. Overall, the work is interesting and novel, and offers a potential strategy for combination therapy in fungal infections that will be of broad interest to the fungal pathogen community.

Response:

Thank you very much for your thoughtful and encouraging feedback on our recent study. We deeply appreciate your thorough examination of our research and are delighted that you found our work to be interesting and novel.

Given the argument that inhibition of the PCA pathway is a good potential therapeutic strategy, it would be interesting to see how well conserved the pharmacological targets (e.g. Acs1) are across the fungal pathogens mentioned in the text (*C. auris*, *C. albicans*, *A. fumigatus*, etc.): Given the argument that inhibition of the PCA pathway is a good potential therapeutic strategy, it would be interesting to see how well conserved the pharmacological targets (e.g. Acs1) are across the fungal pathogens mentioned in the text (*C. auris*, *C. albicans*, *A. fumigatus*, etc.):

Response:

Thank you for your insightful suggestion. We have incorporated sequence homology analysis into the "Discussion" showing that the acetyl CoA synthetases are highly conserved across various fungal pathogens. This by itself is an interesting finding and one with a potential therapeutic impact given the critical role of ACS enzymes in the PCA pathway.

I have significant concerns about the RNAseq analysis. There is no indication how many biological replicates were performed. No QC data are reported for either RNA quality or sequencing quality. The methods specified suggest the analysis was done using an automated pipeline by a non-expert user. I suggest the authors seek support from their core facility or an expert collaborator to ensure their analysis is accurate, that the standard QC parameters are validated, and that their reported methods accurately reflect the rigor of their experimental design. In addition, RNAseq data should be made publically available through a standard repository such as the NCBI SRA.

Response:

We appreciate the reviewer's thorough evaluation and acknowledge the concerns raised regarding the RNAseq analysis. To address these concerns, we have made several modifications to enhance the clarity and rigor of our experiments. Firstly, we have incorporated details regarding the sample designations and the number of replicates (three biological replicates per each *cab1* variant cell type) utilized in our study within the methodology section of the manuscript. Secondly, RNA quality assessment and sequencing were performed by the Yale center of genome analysis (YCGA), a facility known for its expertise in genome research. We have now stated in the manuscript that the RNA integrity number (RIN) values of the analyzed samples ranged from 8.9 to 10, surpassing the minimum required value of 7 for library preparation. We have sought the expertise of specialists to ensure the accuracy and rigor of the analysis and data reporting. As a result, we have thoroughly revised the methodology section. Lastly, the RNAseq data were submitted to the NCBI Sequence Read Archive (SRA).

Fungi is large and diverse kingdom. The authors should specify which organisms they investigate in the abstract.

Response:

: "*Saccharomyces cerevisiae*" is specified in the abstract.

Please note that, if performed as currently described in the methods, all t-tests should be corrected for multiple comparisons, including the RNAseq dataset. The authors should ensure they perform the appropriate tests for the data analyzed throughout the document. Please also note that experiments should be performed in at least biological triplicate, not only technical replicate (see line 462), as is standard practice to control for biological sources of experimental variation. For all figures, the number of biological and technical replicates performed should be indicated, and statistical significance reported in the figure legend. Details of statistical tests should be indicated in the methods section. Where representative images are shown, the number of biological replicates performed should be indicated in the figure legends that accompany the representative image.

Response:

In accordance with standard practice, we have also updated our figures to include clear indications of the number of biological and technical replicates performed, as well as reporting statistical significance in the figure legends.

Line 120-124: Figure 1C is not referred to. Perhaps an error in line 124?

Response:

We apologize for any confusion caused by the discrepancy in figure references. To clarify, the data presented in Fig. 1B and a portion of Fig. 1D pertain to ergosterol drugs, while Fig. 1C and another part of Fig. 1D relate to non-ergosterol antifungal drugs. In order to provide a comprehensive discussion of the ergosterol drug data first, the description originally intended for Fig 1C was moved to after the discussion of Fig 1D and was located in lines 127-135.

Typo line 186- measuring following:

Response:

Thank you for pointing out the error. The text has been updated as follows "was determined by measuring following metabolism of ¹⁴C-pantothenate" has been changed to "assessed by monitoring the phosphorylation and subsequent utilization of ¹⁴C-pantothenate."

Reviewer 2,

In this study, the authors characterize the fungal pantothenate kinase as a target for the potentiation of antifungal drugs. They build on previous knowledge to show that tampering with the activity of the enzyme affect mitochondrial and vacuolar functionality and potentiate the action of various antifungal drugs. The work is well presented, the data is clear, and the experimental approaches are well done. However, in my humble opinion, and despite the high number of experiments performed, I believe this work does not add much new knowledge to the field. In addition, I believe the conclusions claimed by the authors are not sufficiently supported by the results shown.

Response:

We appreciate the careful evaluation of our study and the constructive feedback provided. While we understand the reviewer's perspective on the concerns regarding the novelty and the strength of our conclusions, we respectfully disagree with regard to the novelty of our findings. Our study offers a thorough characterization of fungal pantothenate kinase and the downstream enzymes in the PCA pathway as potential targets for augmenting the effectiveness of antifungal drugs. We contend that our exploration of the pathway's impairment and its consequential effects on vacuolar functionality, leading to increased susceptibility to various antifungal drugs, contributes a valuable breakthrough to the development of novel antifungal therapy. Once again, we appreciate the thoughtful review, and we trust that we have addressed the concerns in this revised manuscript.

To my understanding the point mutation *cab1G351S* causes a temperature dependent phenotype, but the yeast should grow normally at 30°C. In contrast, in Figs. 1E and 2A a defect can be observed in basal growth, despite all growth assays being done at 30°C.

Response:

While the mutant displays a temperature-sensitive growth defect at 37°C, and grows slightly slower than the isogenic strain with a wild type *CAB1* at 30 °C on YPD plates, this strain grows at a rate similar to that of the WT strain in liquid medium at 30°C and inhibition assays are represented as % growth with the growth of each strain is compared in the absence or presence of increasing concentrations of compounds (Fig 1B and C). This is a standard method for determining MIC₅₀ values when comparing different strains and isolates, which might not have a similar growth rate.

The effects of *cab1G351S* and *cab1N290I* on mitochondria and iron homeostasis (and more) have already been published (doi: 10.3390/ijms22010293). The fact that repressing pantothenate kinase activity potentiates antifungal activity was already shown by this manuscript's same authors (doi: 10.1016/j.str.2022.09.001): The effects of *cab1G351S* and *cab1N290I* on mitochondria and iron homeostasis (and more) have already been published (doi: 10.3390/ijms22010293).

Response:

Some of the data have been reported. However, we are now presenting additional data that demonstrate that when the wild-type gene is reintroduced into the mutants, the phenotypes can be restored to those of the WT. Additionally, the data included serve as a starting point for

elucidating the consequential phenotypes that were not reported previously, as illustrated in Fig 2 and 3.

The fact that repressing pantothenate kinase activity potentiates antifungal activity was already shown by this manuscript's same authors (doi: 10.1016/j.str.2022.09.001)

Response:

It is true that previously we reported that inhibition of Pank activity increases amphotericin B susceptibility. In this report, for the first time, we are presenting evidence that inhibition of not only Pank activity but also several steps in the PCA pathway leads to overall drug sensitivity. This sensitivity is not restricted to one specific antifungal drug, such as Amphotericin B, but extends to many different classes of antifungal drugs affecting various targets in fungal cells. Our study provides an explanation for the mechanism behind this regulation by showing that altered Cab1 activity leads to altered vacuolar detoxification.

What evidence is there for a causal link between Cab1 and the vacuole? Although it is of course true that vacuole participate in detoxification, the increased susceptibility to 'xenobiotics' could be caused by other of the pleiotropic effects of tampering with CoA synthesis (like metabolic effects and/or higher ROS, which the authors show and was already known). Thus, the stated in lines 287-288 is not true. The authors should check if the accumulation of any of the stressors inside vacuoles decrease in the cab1 mutants, to prove that its function is impaired. According to Fig. 2B, the integrity of the vacuoles is fine (in contrast to what the authors repeatedly state). The size expansion could be caused by multiple reasons, including defects in growth or oxidative stress.

Response:

We thank the reviewer for their comment. We have now removed the word "integrity" and replaced it with "function" in the title and throughout the manuscript. We have also clarified the text in the manuscript to convey a simple and clear message. Clear evidence has been presented throughout the paper, along with previously reported data, establishing connections between Cab1, CoA, Acetyl-CoA, and the vacuole. The conversion of Acetyl-CoA into HMG-CoA is evidently the first step in the synthesis of ergosterol, a compound crucial for the function of Vacuolar ATPase, a key component of vacuolar function. Furthermore, many erg mutants, deficient in ergosterol production, exhibit similar phenotypes to cab1 mutants, including vacuolar and mitochondrial defects. Our study also demonstrates that Concanamycin A treatment, inhibiting Vacuolar ATPase, results in similar phenotypes of increased susceptibility to antifungal drugs. These phenotypes were evident in the cab1 mutants, as well as in cells with defects in Acetyl-CoA synthesis.

The authors should explain the effects of the point mutations on Pank protein activity.

Response:

We apologize for the confusion. The impact of the point mutations on Pank enzyme activity was delineated in Fig. 4A. To ensure clarity, we have replaced "PA utilization assay using 14C-labeled PA" with "Pank Assay using 14C-labeled PA" in the methodology, and the figure legend. Furthermore, "PA utilization" was replaced by "Pank activity" in the results section.

Given the slight growth defect of the cab1 mutants in normal YPD medium compared with the wild type, the effect of iron and copper seem quite minor. Clearly not 'dramatically reduced' (line 153):

Response:

We removed the word “dramatically” from the text.

The data with PZ-2891 is very confusing. It was described as an activator of pantothenate kinases (in line 257 it is said inhibitor?). The authors state that they do not see an effect on the Cab1 enzymatic activity, but it causes increased accumulation of CoA, which would agree with increased activity. Is the enzymatic assay capable of detecting increase activity? The authors could measure cysteine to detect if the levels are reduced, in line with an increased kinase activity.

Response:

PZ-2891 is an orthosteric activator of human Pank3 enzyme. The compound inhibits hPank3 in the absence of AcCoA but activates it in the presence of AcCoA. However, PZ-2891 has no effect on the yeast Cab1 activity as depicted in Fig. S5. Instead, our findings indicate that PZ-2891 functions as an inhibitor of Acetyl CoA synthetase. The observed increase in cellular CoA levels (Fig. 7A) in PZ-2891 treated cell is a direct result of this inhibition, which cannot be converted to Acetyl-CoA due to the inhibition of Acs by PZ-2891. Our structural modeling also provides evidence that the compound binds to the enzyme, data which have now included in the manuscript.

It was also shown that AR-12 potentiates antifungal action (DOI: 10.1128/AAC.01061-16).

Response:

We agree with the reviewer. Previous studies have shown that AR-12 enhances the efficacy of fluconazole in a mouse model of Cryptococcosis (DOI: 10.1128/AAC.01061-16). However, the underlying mechanism for this enhancement was not fully elucidated. Our findings shed light on the mechanism by which the potentiation mediated by AR2 occurs by linking it to the mechanism by which the PCA pathway modulates vacuolar detoxification and antifungal susceptibility.

(As another question on PZ-2891) It is surprising that PZ-2891, a drug described to modulate human PANK and modelled to bind the fungal enzyme by these authors, have no effect on Cab1 and does on Acs1. Can the authors back the inhibitory action on Acs1 with another enzymatic assay? Given their expertise, can they dock the binding of PZ-2891 to the enzyme? Can they measure cellular Acetyl-CoA to see if it's decreased?

Response:

Indeed, as stated in the above response, PZ-2891 did not exhibit inhibitory effects on the fungal Cab1 enzyme but did inhibit the Acs1 enzyme. In Fig. S5, we conducted fungal Pank enzyme assays in the presence of PZ-2891. While previously characterized fungal Pank inhibitors such as α PanAM and YU385599 effectively inhibited the enzyme activity of both Cab1 WT and mutant proteins, PZ-2891 failed to do so. Conversely, the compound was found to inhibit the Acs protein, and we have included a molecular model of PZ-2891 docking into the AMP binding site of the Acs1 protein, as depicted in Fig. 7C, D, and E.

Reviewer 3,

The manuscript by Choi et al. investigates a connection between vitamin B5 metabolism, vacuolar homeostasis and detoxification of xenobiotics. The authors mainly use genetic methods to show that the PCS pathway is a good target to increase the activity of various xenobiotics. In addition, the authors identify the pantazine analog PZ-2891 as a valid inhibitor of the ACS in yeast. In summary, I think this is a carefully executed study that yields some very interesting findings for the target readership. In summary, I would recommend the manuscript for publication with minor additions. Most of the genetic data are very well executed and largely convincing. However, most of the experiments yield indirect evidence. Could the data be further supported by the analysis of high throughput genetic data from yeast. Multiple genetic interaction studies from yeast are available and are vast sources of data for example the lipid EMAP. Can some of the data be confirmed by measuring the output of the inhibited pathway. For example, could the authors measure ergosterol levels in terbinafine treated cells with and without the potentiating effects?

Response:

Thank you for your insightful comments on our manuscript. We appreciate your positive feedback and suggestions. In our previous report (Chiu et al, doi: 10.1074/jbc.RA119.009791), we conducted a compositional analysis of the components of the ergosterol and its precursors in both the temperature-sensitive *cab1* mutant strain and its wild type counterpart to understand the output of the inhibition of the pathway.

What is the difference between the *Cab1* mutants. I think a better explanation of the three different mutations and their effects on protein function would be useful.

Response:

It seems that different *CAB1* mutations result in different degree of susceptibility toward different classes of anti-fungal drugs and heavy metals. The effects on the protein function were conducted by measurement of the Pantothenate kinase enzyme assay using the proteins harboring the 3 different mutations, The data are included in the Fig. 4.

Improvement in Acs1 Activity Assay: How could the in vitro Acs1 activity assay be improved, and could the authors provide some kinetic data using for example different amounts of substrates?

Response:

The enzyme assay employing various concentrations of inhibitor molecules was done and included in Fig. 7B. Additionally, we conducted molecular structural analysis of Acs1p with PZ-2891, revealing that the compound docks into the AMP binding site of the Acs1 protein as illustrated in Fig. 7C, D, and E. The data suggests that PZ-2891 functions as a competitive inhibitor of the enzyme by competing with AMP for binding to the enzyme, thereby impeding the conformational activation of the enzyme.

We hope you find this revision suitable for publication in Communications Biology.

Best wishes

Choukri Ben Mamoun
Professor of Medicine, Microbial Pathogenesis and Pathology
Yale School of Medicine
New Haven, CT 06520

Reviewers' comments:

Reviewer #1 (Remarks to the Author):

I thank the authors for addressing all concerns.

Reviewer #2 (Remarks to the Author):

The authors have made a great effort to respond to my comments, and I commend them for it. Nevertheless, I believe that there are still two points of concern that need some attention.

1. Growth defects can be displayed on solid or in liquid media, and not always are equal. Spot assays are a classic and effective experiment to determine phenotypes on solid media, as the authors indeed use to report the effects of drugs. According to Figures 1E and 2A is clear that the *cab1G351S* has a slight growth defect on YPD at 30°C. This should be taken into consideration for the other assays, as the potentiation of drugs could partially be caused by this defect.

2. The authors have not demonstrated a causational link between Cab1 and the vacuole. This may not be strictly required, and they do present compelling evidence to support a correlation. However, until a causation is proven, they cannot conclude that "Such broad-spectrum drug susceptibility can only be possible if major mechanisms used by fungi for drug detoxification, such as those mediated by the vacuole, are altered when the PCA pathway is inhibited". There could be other factors involved and the authors should acknowledge this.

Reviewer #3 (Remarks to the Author):

It seems as the authors have only done minor editions to the manuscript and have largely ignored my previous comments:

- The authors did not check any genetic interaction data from yeast which could potentially improve the reliability of the observed data
- The authors point out previous measurements of terbinafine treated cells. However, they did not provide data on the potentiating effect of PZ-2891 on e.g. ergosterol levels
- The question regarding the mutations was aiming at a better description of the mutations. What effect does each mutation have potentially on protein function. What effect is caused by a G351S mutation. Where and in which part of the protein is the glycine 351? Why is a change to serine problematic. Same for the and *cab1N290I* mutant.

I would still like to see these minor points addressed before accepting the manuscript.

Response to Reviewers' Comments

We thank the reviewers for their thorough evaluation of our manuscript. In this revised version, we have diligently addressed all experiments and studies recommended by the reviewers, while also integrating additional analyses to provide supplementary data. Importantly, all generated data align with the conclusions drawn in our original manuscript, reinforcing the overarching message of our work. Below are the point-per-point responses, with reviewers' comments in black and our responses in blue.

Reviewer #2:

The authors have made a great effort to respond to my comments, and I commend them for it. Nevertheless, I believe that there are still two points of concern that need some attention.

1. Growth defects can be displayed on solid or in liquid media, and not always are equal. Spot assays are a classic and effective experiment to determine phenotypes on solid media, as the authors indeed use to report the effects of drugs. According to Figures 1E and 2A is clear that the *cab1*^{G351S} has a slight growth defect on YPD at 30°C. This should be taken into consideration for the other assays, as the potentiation of drugs could partially be caused by this defect.

Response:

We agree with the reviewer's observation and the data provided in Fig. 1 support this observation. However, as we have demonstrated in our studies, inhibition of the PCA pathway, through either genetic mutations or pharmacologically using specific inhibitors of various steps in the pathway, leads to major defects in yeast physiology including alteration of the morphology of the vacuole and its function in drug detoxification. While the *cab1*^{G351S} mutant strain has indeed a slight growth defect (80% that of the WT strain after 48h, consistent with the Figures 1E and 2A) compared to the isogenic parental strain on YPD at 30 °C, the levels of susceptibility of the mutant to various antifungal drugs as quantified using liquid assays with all values normalized to the DMSO (no drug) control were far higher than expected from a 20% decrease in cell growth. Using these analyses, the susceptibility of the *cab1*^{G351S} mutant to antifungals was found to range between 5 and 23-fold that of the isogenic WT strain. Such a dramatic increase in drug susceptibility cannot be accounted for by the slight decrease in growth. Furthermore, this increase in drug susceptibility could be mimicked by inhibiting Cab1 activity or downstream steps in the PCA pathway as demonstrated throughout the manuscript. We believe that our genetic and pharmacological data provide sufficient evidence to support the conclusion of this paper that modulation of the PCA pathway leads to altered susceptibility to antifungal drugs through altered drug detoxification mechanisms, one of which is mediated by the vacuole, an organelle already well known to play a key role in xenobiotic detoxification. We have updated the discussion section of the manuscript to highlight this point.

2. The authors have not demonstrated a causal link between Cab1 and the vacuole. This may not be strictly required, and they do present compelling evidence to support a correlation. However, until a causation is proven, they cannot conclude that "Such broad-spectrum drug susceptibility can only be possible if major mechanisms used by fungi for drug detoxification,

such as those mediated by the vacuole, are altered when the PCA pathway is inhibited". There could be other factors involved and the authors should acknowledge this.

Response:

We thank the reviewer for the insightful comments. We acknowledge the need for caution in interpreting our findings regarding the relationship between the PCA pathway and the vacuole. While we agree that demonstrating a causal link would provide stronger evidence, we believe that our data indeed support a correlation between PCA pathway including Cab1, vacuolar function, and drug susceptibility. We understand that attributing broad-spectrum drug susceptibility solely to alterations in vacuolar function, as indicated by inhibition of the PCA pathway, may oversimplify the complex interplay of cellular mechanisms involved in drug detoxification. Our study aimed to shed light on the role of Cab1/PCA pathway and its potential impact on vacuolar function, but we recognize that other factors affected by the defects in the PCA pathway may also contribute to drug susceptibility.

Given this feedback, we revised the text in the Discussion section of the revised manuscript to acknowledge the possibility of additional factors influencing drug susceptibility beyond alterations in vacuolar function as follows: "In fungi, broad-spectrum susceptibility to drugs and other xenobiotics occurs when drug detoxification mechanisms, such as those mediated by the vacuole are altered."

Reviewer #3:

It seems as the authors have only done minor editions to the manuscript and have largely ignored my previous comments:

1. The authors did not check any genetic interaction data from yeast which could potentially improve the reliability of the observed data

Response:

We thank the reviewer for the suggestion. We have conducted the genetic analysis as recommended and identified genes displaying negative interactions with CAB1 mutations^{1,2}. Subsequently, we conducted functional enrichment analysis of fungal genes using the ShinyGO tool (<http://bioinformatics.sdstate.edu/go/>). Interestingly, our analysis revealed a group of genes (*ATG10*, *ARO2*, *ATG15*, *ATG5*, *ATG7*, *ATG16*, *ATG3*, *ARO7*, and *ARO1*) involved in the autophagy pathway. The autophagy pathway is activated under nutrient-deprived conditions, leading to the formation of autophagosomes. These autophagosomes ultimately fuse with the vacuole, where nutrients are digested and recycled. Notably, genes implicated in autophagosome fusion with vacuole, such as *VPS39*, *YPT7*, *YKT6*, and *VAM7*, were also found showing genetic interactions with the *cab1* mutation. The exacerbation of growth phenotypes observed when genes involved in autophagosome formation and vacuolar fusion were combined with the *cab1* malfunction strongly supports our hypothesis regarding the impact of *cab1* mutation and PCA pathway targeting on vacuolar function. We have updated

the Discussion section of the manuscript to include this information and included a new figure Fig. S8 in the Supplementary Data section of the manuscript to highlight these findings.

2. The authors point out previous measurements of terbinafine treated cells. However, they did not provide data on the potentiating effect of PZ-2891 on e.g. ergosterol levels.

Response:

We thank the reviewer for the suggestion and agree that determining ergosterol levels upon treatment with terbinafine and PZ-2891, both of which target specific steps upstream of ergosterol synthesis, could generate useful information. Our previous studies have already shown that reduced Cab1 activity results in reduced levels of lanosterol and other intermediates in the ergosterol biosynthesis, and more recent studies (not shown) found that HMG-CoA (a critical precursor for ergosterol biosynthesis) is also significantly reduced in mutants with reduced Cab1 activity. Based on all the genetic and pharmacological data we have generated so far, we expect that potentiators such as PZ-2891 or AR-12 will have a similar effect. With regards to ergosterol levels, most of the reported studies aimed at measuring ergosterol levels focused on the steady state levels of this sterol. The ideal studies are those that could monitor the de novo synthesis of ergosterol from pantothenic acid. However, all these studies are time consuming (at least 3 months of LC and GC-MS analyses) and cost involved (>\$5,000). Furthermore, we believe that conducting experiments to measure ergosterol levels in the presence of the potentiator may not significantly add to the main conclusion of our paper, which demonstrates the role of the PCA in the regulation of drug detoxification, and the translation of this knowledge to develop a new therapeutic strategy based on the use of potentiators to fight multidrug resistance. We also would like to add that even if we demonstrate decreased levels of newly synthesized ergosterol, it will still remain unclear how ergosterol regulates vacuole V-ATPase or other vacuolar functions leading to drug detoxification. We would greatly appreciate it if the assay could be reserved for future studies focused primarily on unravelling the mechanism by which the PCA pathway regulates drug detoxification. As mentioned above, genetic interactions have identified genes involved in autophagy, vesicular fusion and degradation of inner vesicles with the vacuole to be involved in negative genetic interactions with a *cab1* mutant. These findings indicate that there are several other possible mechanisms by which the PCA could regulate vacuolar detoxification. The editor agrees it is not necessary for these data to be added to the manuscript. Unravelling such mechanisms in future studies is warranted considering the important of the PCA pathway in the regulation of fungal susceptibility to antifungal drugs. We have updated the Discussion section of the manuscript to comment on these points and added a new Fig. S8 to highlight these data.

As mentioned above, genetic interactions have identified genes involved in autophagy, vesicular fusion and degradation of inner vesicles with the vacuole to be involved in negative genetic interactions with a *cab1* mutant. These findings indicate that there are several other possible mechanisms by which the PCA could regulate vacuolar detoxification. Unravelling such mechanisms in future studies is warranted considering the important of the PCA pathway in the regulation of fungal susceptibility to antifungal drugs. We have updated the Discussion section of the manuscript to comment on these points.

3. The question regarding the mutations was aiming at a better description of the mutations. What effect does each mutation have potentially on protein function. What effect is caused by a G351S mutation. Where and in which part of the protein is the glycine 351? Why is a change to serine problematic. Same for the and cab1N290I mutant. I would still like to see these minor points addressed before accepting the manuscript.

Response:

We appreciate the thorough inquiries regarding the *cab1* mutations and their functional and structural implication. As previously described by Gihaz et al.³ and Chiu et al.⁴, the G351S mutation results in a dramatic reduction in *cab1* enzymatic activity, with a higher K_m and a lower V_{max} compared to WT *Cab1*. These findings were also validated in this manuscript as *cab1*^{G351S} yeast mutant exhibited lower PanK cellular activity compared to *cab1* WT (Fig. 4A and Figure S5). This highly conserved residue (Olzhausen et al. 2009,⁵) is likely to play a critical role in the conformational changes that occur upon binding to PA.

Based on the crystal structure of *cab1*, which was solved at 1.8 Å, Gly351 is positioned on a helical loop located 9-10 Å from the catalytic Asp24 and Glu105 and 15 Å from the catalytic Tyr220 of the adjacent monomer³. Given that the structure of *cab1*^{G351S} mutant has not yet been solved, we cannot determine the exact effect of this site-specific mutation on the enzyme structure. Nevertheless, residue Gly351 participates in a hydrogen bonds network together with structurally conserved water molecule and His 119. Serine mutation might disrupt this interaction leading to altered folding around this region. Another possibility is that position Gly351 is partially buried in the enzyme core. In such a location, introducing a larger moiety compared to H atom (glycine side chain) could cause a steric clash with neighbor residues which might lead to less optimal enzyme efficiency.

As was reported by Ceccatelli Berti et al.⁶, *cab1* mutant N290I was designed to mimic the mutation N500I found in the human PANK2 gene and associated with pantothenate kinase-associated neurodegeneration (PKAN). The missense mutation N500I has been associated with late onset of the PKAN disease and caused significant loss of PANK2 enzyme activity^{7,8}. Similar to the G351S mutation, the N290I mutation led to a major alteration of *cab1* function in yeast cells with phenotypes including thermosensitivity, sensitivity to metals, low oxygen

consumption, and inability to utilize non-fermentable carbon sources^{6,9}. Like its corresponding residue N500 in PANK2, position N290 is located in the dimer interface of the Cab1 structure³. This charged and highly-conserved residue participate in intermolecular hydrogen bonds between Cab1 monomers. Since the phosphorylation of PA depends on initial proton cleavage by Glu105, which is stabilized by the catalytic Tyr220 from the neighbor chain, dimerization plays a crucial role in enzyme catalysis¹⁰. It was also shown that cab1 mutant N290A exhibited altered phosphorylation activity³. Thus, destabilization of the hydrogen bond network within the Cab1 interface (Ala or Ile mutations) can result in reduced activity, kinetic parameters, and proper functionality in yeast cells^{3,9}.

We added a new section in the Introduction section of the manuscript to highlight this information.

We hope you find this revision suitable for publication in Communications Biology.

Best wishes

Choukri Ben Mamoun
Professor of Medicine, Microbial Pathogenesis and Pathology
Yale School of Medicine
New Haven, CT 06520

References

- 1 Costanzo, M. *et al.* The genetic landscape of a cell. *Science* **327**, 425-431 (2010). <https://doi.org:10.1126/science.1180823>
- 2 Costanzo, M. *et al.* A global genetic interaction network maps a wiring diagram of cellular function. *Science* **353** (2016). <https://doi.org:10.1126/science.aaf1420>
- 3 Gihaz, S. *et al.* High-resolution crystal structure and chemical screening reveal pantothenate kinase as a new target for antifungal development. *Structure* **30**, 1494-1507 e1496 (2022). <https://doi.org:10.1016/j.str.2022.09.001>
- 4 Chiu, J. E. *et al.* The antimalarial activity of the pantothenamide alpha-PanAm is via inhibition of pantothenate phosphorylation. *Sci Rep* **7**, 14234 (2017). <https://doi.org:10.1038/s41598-017-14074-9>
- 5 Olzhausen, J., Schubbe, S. & Schuller, H. J. Genetic analysis of coenzyme A biosynthesis in the yeast *Saccharomyces cerevisiae*: identification of a conditional mutation in the pantothenate kinase gene CAB1. *Curr Genet* **55**, 163-173 (2009). <https://doi.org:10.1007/s00294-009-0234-1>
- 6 Ceccatelli Berti, C., Gilea, A. I., De Gregorio, M. A. & Goffrini, P. Exploring Yeast as a Study Model of Pantothenate Kinase-Associated Neurodegeneration and for the Identification of Therapeutic Compounds. *Int J Mol Sci* **22** (2020). <https://doi.org:10.3390/ijms22010293>
- 7 Zhou, B. *et al.* A novel pantothenate kinase gene (PANK2) is defective in Hallervorden-Spatz syndrome. *Nat Genet* **28**, 345-349 (2001). <https://doi.org:10.1038/ng572>
- 8 Zhang, Y. M., Rock, C. O. & Jackowski, S. Biochemical properties of human pantothenate kinase 2 isoforms and mutations linked to pantothenate kinase-associated neurodegeneration. *J Biol Chem* **281**, 107-114 (2006). <https://doi.org:10.1074/jbc.M508825200>

- 9 Ceccatelli Berti, C. *et al.* Evidence for a Conserved Function of Eukaryotic Pantothenate Kinases in the Regulation of Mitochondrial Homeostasis and Oxidative Stress. *Int J Mol Sci* **24** (2022). <https://doi.org:10.3390/ijms24010435>
- 10 Subramanian, C. *et al.* Allosteric Regulation of Mammalian Pantothenate Kinase. *J Biol Chem* **291**, 22302-22314 (2016). <https://doi.org:10.1074/jbc.M116.748061>

REVIEWERS' COMMENTS:

Reviewer #2 (Remarks to the Author):

All my concerns and suggestions have been addressed

Reviewer #3 (Remarks to the Author):

The authors have addressed my concerns. I still disagree with the authors that the ergosterol measurements would be useful but I understand the time and money concerns. Therefore I think the manuscript can be accepted in its current form.